# COCO-Periph:
# Bridging the Gap Between Human and Machine Perception in the Periphery

**Anne Harrington**[1,2]   **Vasha DuTell**[1,2]   **Mark Hamilton**[1]   **Ayush Tewari**[1]
**Simon Stent**[3]   **William T. Freeman**[1]   **Ruth Rosenholtz**[1,2]
[1]MIT CSAIL   [2]MIT Brain and Cognitive Sciences   [3] Toyota Research Institute
{annekh,vasha}@mit.edu

## Abstract

Evaluating deep neural networks (DNNs) as models of human perception has given rich insights into both human visual processing and representational properties of DNNs. We extend this work by analyzing how well DNNs perform compared to humans when constrained by peripheral vision – which limits human performance on a variety of tasks, but also benefits the visual system significantly. We evaluate this by (1) modifying the texture tiling model (TTM), a well tested model of peripheral vision, to be more flexibly used with DNNs, (2) generating a large dataset which we call COCO-Periph that contains images transformed to capture the information available in human peripheral vision, and (3) comparing DNNs to humans at peripheral object detection using a psychophysics experiment. Our results show that common DNNs underperform at object detection compared to humans when simulating peripheral vision with TTM. Training on COCO-Periph begins to reduce the gap between human and DNN performance and leads to small increases in corruption robustness, but DNNs still struggle to capture human-like sensitivity to peripheral clutter. Our work brings us closer to accurately modeling human vision, and paves the way for DNNs to mimic and sometimes benefit from properties of human visual processing.

## 1   Introduction

Deep neural networks (DNNs) have shown great promise as models of human visual perception, enabling the prediction of both neural response patterns (Yamins et al., 2014; Rajalingham et al., 2015; Yamins & DiCarlo, 2016; Kell & McDermott, 2019) and aspects of visual task performance (Yamins et al., 2014; Geirhos et al., 2018; Mehrer et al., 2021). However, there are still critical differences in how computer vision DNNs process information compared to humans (Rajalingham et al., 2015; Geirhos et al., 2020; Wichmann & Geirhos, 2023). These differences are evident in psychophysical experiments (Berardino et al., 2017; Feather et al., 2019; Hénaff et al., 2019; Harrington et al., 2022) and adversarial examples (Szegedy et al., 2013; Elsayed et al., 2018; Ilyas et al., 2019). One difference between DNNs and humans that has gained recent interest is the existence of peripheral vision in humans. Peripheral vision describes the process in which human vision represents the world with decreasing fidelity at greater eccentricities, i.e. farther from the point of fixation. Over $99\%$ of the human visual field is represented by peripheral vision. While it is thought to be a mechanism for dealing with capacity limits from the size of the optic nerve and visual cortex, peripheral vision has also been shown to serve as a critical determinant of human performance for a wide range of visual tasks (Whitney & Levi, 2011; Rosenholtz, 2016).

The benefits of modeling peripheral vision in DNNs are two-fold. For applications that require predicting or mimicking human performance on a visual task – like predicting if a driver will detect a hazard – DNNs in computer vision must capture aspects of human peripheral vision that drive task performance. For applications in representation learning, peripheral vision represents a biological strategy that presumably evolved to efficiently and robustly solve a variety of tasks in spite of information loss due to significant constraints on the system. DNNs might benefit from these representational strategies in areas such as robustness, where a link between adversarial robustness and human visual representations has already been seen (Engstrom et al., 2019; Ilyas et al., 2019; Harrington & Deza, 2022).

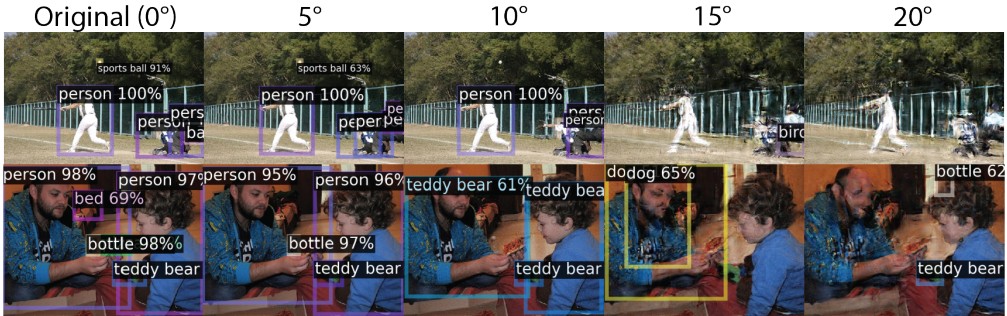

Figure 1: **The COCO-Periph Dataset** contains MS-COCO images that have been transformed to visualize the loss of information in human peripheral vision due to visual crowding (not just acuity loss) at various eccentricities. This loss of visual information causes difficulties for computer vision systems: models such as Faster-RCNN (shown here) perform poorly at tasks like object detection as simulated eccentricity increases. In contrast, human performance is known to fail gracefully, raising the question: how can we close this gap?

Accurately modeling peripheral vision in DNNs, however, is challenging. Current DNN approaches are disjoint and a number of them require specialized architectures (Jonnalagadda et al., 2021; Min et al., 2022), only model a loss of resolution (Pramod et al., 2022) – which is insufficient to predict effects of peripheral vision like crowding (Balas et al., 2009), or rely on style transfer approaches (Deza & Konkle, 2020) which are not as well tested as statistical models. In human vision science, peripheral vision has been well-modeled with multi-scale pyramid-based image transformations that, rather than predicting performance on a particular task, instead output images transformed to represent the information available in peripheral vision. Humans viewing these transformed images perform visual tasks with an accuracy that well predicts human performance while fixating the original images (Ehinger & Rosenholtz, 2016; Rosenholtz et al., 2012b; Freeman & Simoncelli, 2011).

In this work, we leverage one of these pyramid-based peripheral vision models, the Texture Tiling Model (TTM) (Rosenholtz et al., 2012b), to simulate peripheral vision in a variety of DNN models. We do so by modifying TTM to use a uniform rather than a foveated pooling operation (uniform-TTM); this allows us to model a single point in the periphery without having to choose a fixation. We use uniform TTM to render a popular object dataset, MS-COCO (Lin et al., 2014), to simulate peripheral vision at the input level for DNNs – we call the transformed dataset COCO-Periph. To understand the effect that peripheral vision-like inputs have on DNN performance, we perform a human psychophysics experiment measuring object detection in the periphery, and then design a machine psychophysics experiment to test DNNs on the same task. We compare detection results between humans and DNNs and show a gap in performance between the two. This gap can be reduced by training on COCO-Periph, but we still see noticeable differences in sensitivity to clutter.

The COCO-Periph dataset is one of the largest datasets for studying peripheral vision in DNNs, and our analysis represents one of the broadest evaluations of peripheral vision in modern DNNs to date. COCO-Periph itself is a significant contribution, representing over 6 months of compute time that makes it computationally feasible to test TTM in DNNs and standardizes the evaluation of peripheral vision in computer vision. We publicly release our COCO-Periph dataset, along with code for uniform TTM and the psychophysics analyses at https://github.com/RosenholtzLab/COCOPeriph to enable further research into human and machine perception in the periphery – paving the way for DNNs to mimic and benefit from properties of human visual processing.

## 2 BACKGROUND AND RELATED WORK

### 2.1 PERIPHERAL VISION

Often misunderstood as a simple loss of acuity, peripheral vision in reality involves much more complex processes. While the retina does display a progressive reduction of photoreceptor density as a function of eccentricity, most of the information loss in peripheral vision occurs downstream

in the visual cortex. The phenomenon of visual crowding exemplifies this where human peripheral performance degrades in the face of clutter from the spacing of nearby objects and the features of local image regions (Vater et al., 2022).

Peripheral vision has been successfully modeled as a loss of information in representation space (Rosenholtz et al., 2012b; Freeman & Simoncelli, 2011), where models like TTM (Rosenholtz et al., 2012b; Rosenholtz, 2020) perform a texture-processing-like computation of local summary statistics within pooling regions that grow with eccentricity and tile the visual field. TTM relies on the Portilla and Simoncelli statistic set (Portilla & Simoncelli, 2000), very similar to the Freeman and Simoncelli model (Freeman & Simoncelli, 2011). Some more recent models evaluate different statistic and show strong performance on metameric tasks (Deza et al., 2017; Wallis et al., 2019; Broderick et al., 2023). Among these, TTM is one of the only to be validated against human performance on an extensive number of behavioral tasks, including peripheral object recognition, visual search, and a variety of scene perception tasks (Ehinger & Rosenholtz, 2016). Although TTM is powerful, the computational requirements of synthesizing TTM transforms make it impractical to use online at the large scale of DNNs. Synthesizing a single TTM transform image can take 5+ hours. This has been addressed in part by (Brown et al., 2021), which modified the optimization process for transform generation with gradient descent, allowing GPU-optimization, and (Deza et al., 2017) and (Wallis et al., 2017) which incorporated style-transfer into the procedure. However, these models are not as well validated on human performance as TTM, and most are still not fast enough to use during DNN training. To facilitate large experiments, we create COCO-Periph – a large-scale dataset that pre-computes these images with a more flexible fixation scheme.

## 2.2 Human-Inspired Deep Neural Networks

Extensive work has been done in creating biologically-inspired object recognition models. A number of these models have been shown to impart representational benefits such as robustness to occlusion (Deza & Konkle, 2020), generalization across scale (Zhang et al., 2019; Han et al., 2020), and adversarial robustness (Vuyyuru et al., 2020; Dapello et al., 2021; Guo et al., 2022). It has also been suggested that adversarial training alone can improve human perceptual alignment (Dapello et al., 2020; Feather et al., 2022; Ilyas et al., 2019; Harrington & Deza, 2022). Research in this domain overall has greatly benefited from DNN benchmarks such as BrainScore (Schrimpf et al., 2020) that compare models to humans using neural and behavioral data.

Despite clear benefits on recognition tasks, modeling human vision is less explored in more complex tasks like object detection. One exception to this includes FoveaBox which takes inspiration from foveation in human vision to simultaneously predict object position and boundaries without anchors (Kong et al., 2020). Additionally, training on a stylized version of COCO (Michaelis et al., 2019) (much like the stylized ImageNet work which reduced texture bias and increased shape bias in recognition models (Geirhos et al., 2018)) was shown to increase corruption robustness in object detection DNNs. Peripheral vision, however, is thought to use texture-like representations, and is critically involved in tasks where context matters like detection. Testing peripheral vision in tasks like detection is key to understanding the benefits and trade-offs of modeling human vision in DNNs.

## 3 Uniform Texture Tiling Model

To model critical aspects of peripheral vision without assuming a fixation point, we use a modified version of TTM that relies on uniform, rather than foveated pooling. In original TTM (Rosenholtz, 2020), pooling regions grow linearly with eccentricity, but for uniform TTM, we fix the pooling region size to match the ring of pooling regions at a single eccentricity (see Appendix Sec. A.1). For example, we set the pooling region size to correspond to $15°$ eccentricity, as in Figure 2. With that uniform pooling, we can create images that show the information available as if each pooling region appeared at the same eccentricity. Though this represents an impossible situation, it provides two practical advantages: (1) the ability to shift the modeled fixation by stitching together pre-computed uniformly transformed images to create pseudo-foveated images (see Appendix Sec. A.2), and (2) to evaluate both human and machine performance for an entire image at a single eccentricity.

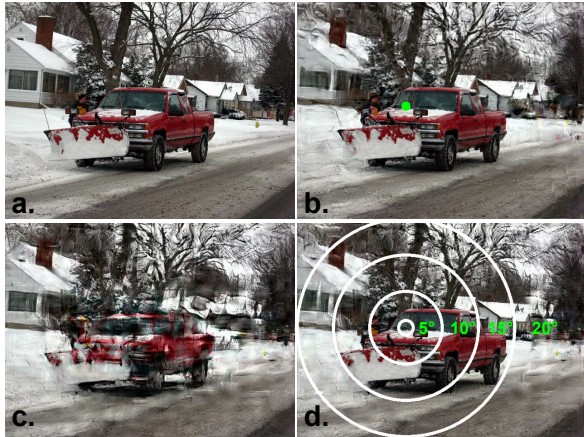

Figure 2: **Original vs. Uniform Texture Tiling Model (TTM).** An original image (a) can be processed using the Original TTM (b) which is foveated, meaning that information is pooled in regions that grow farther from an assumed fixation point (green dot). To create our dataset we adapt TTM to use a fixed uniform pooling region size everywhere in the image, shown in (c) at $15°$ eccentricity. We can efficiently stitch together multiple uniform TTM images from our pre-computed dataset to recreate the foveated effect (d).

## 4   COCO-PERIPH DATASET

We apply Uniform TTM to the COCO dataset, creating COCO-Periph which contains images transformed like peripheral vision. COCO-Periph allows us to use the highly tested Texture Tiling as an input pre-processing step to train and evaluate DNNs. In COCO-Periph, we rendered images that capture the amount of information available to humans at $(5°, 10°, 15°, 20°)$ in the periphery, assuming 16 pixels per degree. (For reference, the width of a full-screen image on a laptop at a typical viewing distance subtends $20° − 40°$). COCO-Periph contains the entire COCO 2017 validation and test set, and over 74K, 117K, 118K, and 97K of the 118K total training images transformed to model $5°, 10°, 15°$, and $20°$ of eccentricity, respectively.

Measuring object detection performance on COCO-Periph using the original COCO ground truth labels, we see in Table 1 for a variety of pre-trained models that average precision (AP) degrades with distance in the periphery. De-noising models, which have the highest baseline scores, perform the best overall compared to the other architectures measured. Performance likely degrades because COCO-Periph is potentially out of distribution for models, and at farther eccentricities, objects have a greater potential to move due to the larger pooling regions used in uniform TTM. To understand how the degradation in performance we see compares to human vision, we conduct a psychophysics experiment in Section 5. In the psychophysics analysis, we address box localization issues (see Sec 6) and train a Faster-RCNN on COCO-Periph (See Table 1 bottom row and Sec. 7.1).

## 5   HUMAN PSYCHOPHYSICS: OBJECT DETECTION IN THE PERIPHERY

To compare DNN performance to humans in the periphery, we first collected human psychophysics data on an object detection task. We choose a detection rather than a recognition task because humans can guess object identity quite well based on context alone, i.e. even when the object itself is occluded (Wijntjes & Rosenholtz, 2018). In our detection task, we present two images on every trial, identical except for the presence or absence of a particular object, and ask a human subject to judge which of the two images contained a target object. For the object present images, we choose 26 images from the COCO validation set that have one instance of an object. For the absent image, we remove that object via in-painting (see Appendix Sec. A.3.1). We selected images with a variety of small to medium sized objects in different scenes.

In each trial, 10 eye-tracked subjects fixated at a specified location either $5°, 10°, 15°$, or $20°$ away from the target object, and viewed an object present and absent image in random order. Subjects

| Model Arch | AP 0° | AP 5° | AP 10° | AP 15° | AP 20° |
|---|---|---|---|---|---|
| DINO-FocalNet-Large (Zhang et al., 2022) | **58.4** | **51.6** | **44.4** | **20.2** | **15.0** |
| DINO-Swin-Tiny (Zhang et al., 2022) | 51.3 | 44.0 | 34.1 | 11.1 | 7.6 |
| Detr-R50 (Carion et al., 2020) | 42.0 | 35.2 | 25.1 | 6.9 | 4.5 |
| RetinaNet-R50 (Lin et al., 2017) | 38.7 | 31.5 | 22.1 | 6.9 | 5.0 |
| FoveaBox (Kong et al., 2020) | 40.4 | 33.2 | 23.4 | 7.5 | 5.3 |
| Faster-RCNN-X101 (Ren et al., 2015) | 39.6 | 32.6 | 21.8 | 6.6 | 4.7 |
| Faster-RCNN-R50 (Ren et al., 2015) | 36.7 | 29.4 | 19.9 | 5.9 | 4.2 |
| All ° Train Faster-RCNN-R50 | 33.8 | 30.5 | 28.1 | 15.8 | 12.7 |
| All ° FT Faster-RCNN-R50 | 36.1 | 31.8 | 27.7 | 13.9 | 10.8 |

Table 1: **Average Precision (AP) on COCO and COCO-Periph Validation Set**. 0° refers to performance on unchanged MS-COCO data. The other AP values correspond to different eccentricities of uniform TTM transform images.

were asked to report which image contained the specified object in a two-interval-forced-choice paradigm (2IFC), viewing 10 present/absent image pairs at each eccentricity. As a control, we also tested subjects on the same task but on uniform TTM transformed images. Details on these experimental setups can be found in the Appendix Sec. A.3.2, A.3.3, and A.4.

We find overall that human object detection performance always degrades progressively with increasing eccentricity (Figure 3, blue line). Detection ability is consistently strong at 5°. However, for some images observers reach near chance performance at 20° eccentricity, whereas a few image pairs have objects that are easily detected at all eccentricities. Often, high color contrast between the object and its background and a lack of clutter from other nearby objects made target objects more easily detected in the periphery, leading to better performance, which is consistent with the crowding literature. See Appendix Sec. A.4.1 for per-image human accuracy.

## 6 MACHINE PSYCHOPHYSICS

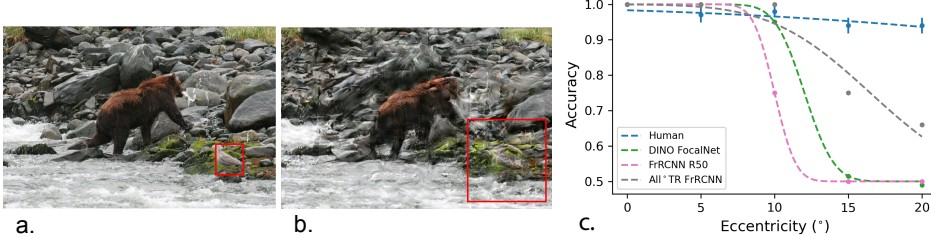

a.        b.        c.

Figure 3: **Example Easy Object Detection.** **(a)** Original image with target object bounding box, and **(b)** TTM transform for 15° (240 pixels) with extended bounding box (used to perform machine object detection task). **(c)** Human accuracy for peripheral viewing of example image averaged over all subjects with error bars reporting SEM (blue), compared with accuracy on TTM image for a pretrained (pink) and trained on COCO-Periph (gray) Faster RCNN R50 model, and a DINO FocalNet model (green). Psychometric curves are fit with an inverse cumulative normal distribution.

To compare human and DNN performance, we have DNN object detection models perform the same two-alternative/interval forced choice task given to human subjects. We do this by first using uniform TTM to generate 10 different peripheral transform images for each of the object present and absent images, at each of the 4 tested eccentricities (5°, 10°, 15°, 20°). Because TTM is a stochastic model that is under-constrained compared to image pixel values, each of the 10 TTM transform images differ from one another. This gives us 100 unique present/absent transform pairs for each original image/eccentricity combination. For each pairing, we input a transformed image to the object detection model with low detection threshold (0.01) to get proposed bounding boxes and object scores. We then determine if the proposed box overlaps with a padded ground truth box of the target object; we pad the ground truth box by half the width of a pooling region to account for

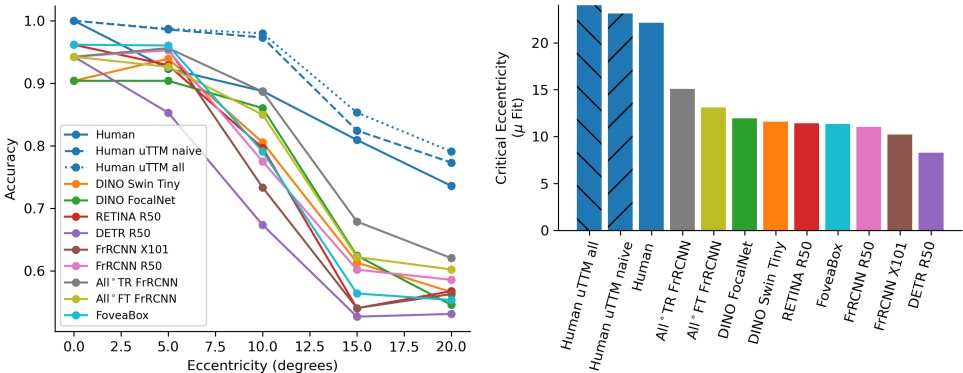

Figure 4: **Detection Accuracy and Critical Eccentricity for Humans and DNNs averaged over all images.** Accuracy averaged across images and over subjects for human data is reported by eccentricity (**left**). We show data for humans viewing original images (Human), and humans viewing uniform TTM (uTTM) images (hatched blue bars: 3 naive subjects, 5 all subjects). To this data, we fit psychometric functions. We summarize performance across eccentricity as the critical fall-off point $\mu$ (**right**). In both plots, we observe a noticeable drop in performance near large eccentricities.

position uncertainty introduced in human peripheral vision and TTM (see Figure 3 a and b for an example of $15°$ padding). To measure how strongly the DNN predicts there is an object in padded box region, we sum the total scores of all objects (regardless of predicted class) that overlap at least 75 percent (intersection of area with respect to the proposed box). We score the model as correct on a trial if the total object scores for the present image are greater than the absent. We score incorrect if the absent is greater and give a half score if present and absent are equal. We take the average over all the present-absent pairing accuracies for each eccentricity. See Appendix Sec. A.5 for pseudo-code and Figure 14 for the general workflow.

To keep the comparison between DNNs and humans fair, we do not enforce that the model must predict the correct object identity when scoring predictions at each trial. Because we use a forced-choice paradigm, humans subjects can give a correct response by simply detecting the presence of any object at the approximate right location, rather the specified one. Although we specify an object class to human subjects, this strategy is likely to happen when peripheral information is poor.

# 7    HUMAN VS MACHINE PERFORMANCE AT PERIPHERAL OBJECT DETECTION

Like the human observers, DNNs' response accuracies are highly image-dependent, with some pairs resulting in poor performance for all models. While human performance falls gradually for most images, DNN object detectors can often retain good accuracy for the $5°$ eccentricity TTM transforms, but many show sharp falloffs in accuracy to chance performance soon after (See Figure 3 for a representative example).

To quantitatively compare performance, we fit both human and DNN performance data across eccentricity to a psychometric function for each image. We use a reverse cumulative Gaussian distribution which determines the critical (75% correct, halfway between perfect and chance performance) eccentricity by the mean of the distribution ($\mu$), and the performance falloff rate by ($\sigma$) (Strasburger, 2020).

For all images tested, humans outperform all object detection models, with critical thresholds more than $5°$ greater than detection models (Figure 4). We find generally weak correlations between between DNN and human performance for critical eccentricity ($\mu$) (see Appendix Fig. 17). Among the pre-trained models, DINO detectors have the closest critical eccentricity to humans and have the strongest correlation.

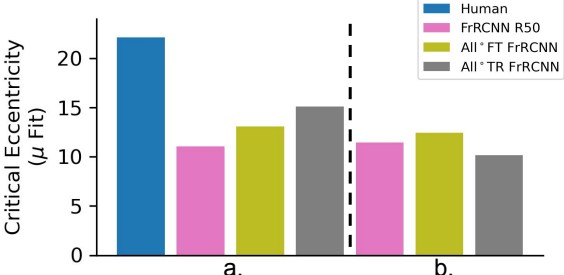

Figure 5: **Detection vs Recognition.** **(a)** Fine-tuning and training on COCO-Periph increases baseline Faster-RCNN-R50 performance at the object detection psychophysics task. **(b)** Machine psychophysics performance on recognition. We pass the classification head of Faster-RCNN models the groundtruth bounding box and score performance based on recognition in that region. Data is averaged over all experiment images.

## 7.1 TRAINING ON COCO-PERIPH

To reduce the gap between human and DNN performance, we fine-tune and train a ResNet-50 backbone Faster-RCNN-FPN detection model on COCO-Periph (See Appendix Sec. A.8 for AP results and training details, and Appendix Fig. 22 for bounding box examples). For fine-tuning (plotted as All° FT RCNN) we start from detectron2's (Wu et al., 2019) pre-trained model and use a 3x iteration scheme with a lowered learning rate. When training from scratch (plotted as All° FT RCNN), we use the default 3x training configuration in detectron2.

We find that training a model from scratch with all eccentricities in COCO-Periph plus original COCO images (0°) produces the best performing model in the psychophysics evaluation (see Figure 4). The model trained with COCO-Periph has a critical eccentricity of nearly 5° greater than the pre-trained baseline (Figure 4, 5.a) The fine-tuned model, however, under-performs the trained model which we suspect is because of the lowered learning rate during training and a decrease in baseline average precision. In addition, we report that DNN psychophysics performance is similar on uniform, original TTM, and pseudofoveated TTM (see Appendix Fig. 15, 16).

To better understand the impact training on COCO-Periph has on the psychophysics performance, we additionally evaluate object recognition in the machine psychophysics (Figure 5). We give the classification head of Faster-RCNN-R50 models the padded ground truth bounding box of the target object. We then score models based on which image, present or absent, has the highest classification probability for the target object. Unlike the detection version of the task, we find that training from scratch performs worse than baseline. This could indicate the trained model improved more at localizing objects rather identifying them in the periphery.

| Model | $mAP_c$ | severity= 1 | severity= 5 | brightness | elastic transform |
|-------|---------|-------------|-------------|------------|-------------------|
| Faster-RCNN | 17.34 | **25.08** | 9.57 | **28.29** | 11.94 |
| All ° FineTune | **17.47** | 24.80 | 9.84 | 27.31 | 14.48 |
| All ° Train | 16.72 | 23.28 | **9.88** | 25.31 | **15.02** |

Table 2: **Corruption Robustness Average Precision ($mAP_c$) on COCO Validation Set**. All models are Faster-RCNN ResNet50 FPN architecture (Ren et al., 2015). ($mAP_c$) refers to AP over 15 corruptions at 5 severity levels (Michaelis et al., 2019). Columns (severity= 1) and (severity= 5) report AP over all corruptions at a single severity level. Last two columns (brightness, elastic) report AP for the worst and best performing corruption compared to baseline.

In order to see if there were benefits from training on COCO-Periph beyond getting closer to human performance, we evaluated trained models for corruption robustness using the COCO-C suite of corruptions (Michaelis et al., 2019). We find that corruption robustness improves most noticeably for geometric transformations like 'glass blur' and 'elastic transform'. Interestingly, performance is slightly lower than baseline for noise-like corruptions and ones that change contrast (See Appendix Fig. 26). Averaged over all corruptions, average precision is slightly higher for the model trained from scratch on COCO-Periph than baseline.

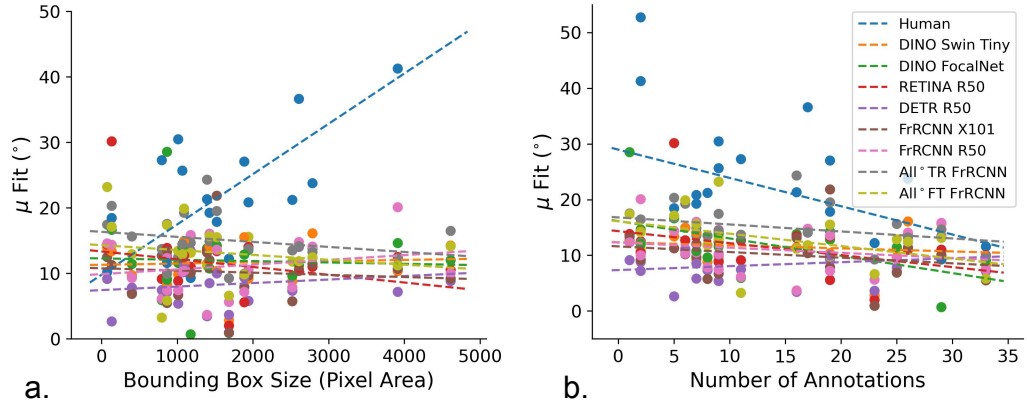

Figure 6: **Object Size and Clutter. a:** Object size predicts critical eccentricity for human observers (blue), but remains low for large objects in all object detection models tested. **b:** In humans (blue), critical eccentricity is highest for images with few objects, with performance decreasing as images become more crowded with objects. This relationship is not observed for tested object detection models, where critical eccentricities remain low.

## 7.2 EFFECTS OF OBJECT SIZE AND CLUTTER

Since both human and DNN performance strongly varied by image, we looked for image properties that might predict performance, and asked if these had similar effects for humans and computer vision DNNs. Examining critical eccentricity as a function of object size, humans have a higher critical eccentricity for larger objects; that is, human performance increases with progressively larger target objects (Figure 6). Surprisingly, this relationship does not appear to hold for any object detection model, even the ones trained on COCO-Periph which have higher AP-large and AP-small than baseline on TTM transform images (see Appendix Sec. A.8.1).

Human object detection performance in the periphery is known to be strongly mediated by the amount of clutter. One measure of clutter is the number of objects near the target. To test if this holds true for object detection models in our experiment, we used the number of ground truth COCO annotations in the image as a proxy for clutter (note clutter can be present in specific sub-regions of an image, and that COCO annotations do not label all objects in many scenes). As expected, human performance decreases as images become more cluttered (Figure 6). Performance in object detection models does not show a strong relationship with clutter. Interestingly, this is true even for models trained on COCO-Periph, which should reflect the degrading effect of clutter on the peripheral representation, according to TTM.

## 8 DISCUSSION

To evaluate the effects of peripheral vision on DNNs, we modified a summary statistic model of peripheral vision (TTM) to be more flexibly pre-computed as an image transformation. We then rendered a large subset of the COCO dataset (COCO-Periph) to model 5°, 10°, 15°, and 20° degrees in the periphery to feed into object detection DNNs. To compare performance against humans, we collected human data on detection in the periphery and measured performance against DNNs.

### 8.1 UNDERSTANDING DIFFERENCES IN HUMAN AND DNN PERFORMANCE

Our results expose a gap in performance between humans and computer vision DNNs in the periphery. When we restrict DNNs to input matching human peripheral vision, detection performance matches humans for some models at low eccentricities, but quickly becomes brittle, degrading sharply with eccentricity, whereas human performance falls off smoothly. *What underlies the noted differences in performance and could explain this? Can we rule out limitations of TTM itself?* While TTM is widely-tested, like all other peripheral vision models, it sometimes under-predicts human performance, and validation has primarily been in greyscale images (although TTM-like models have been tested for color under the metamerism task (Brown et al., 2021), and see (Jagadeesh & Gardner, 2022)). We argue that while these limitations of TTM may put a ceiling on model perfor-

mance, TTM uses the same statistics at all eccentricities, which cannot explain the sharp falloff in performance at high eccentricities as compared to humans. This along with differences in performance between models indicate that gaps between human and machine performance are unlikely to be explained by stimuli alone. Furthermore, we note multiple aspects in which we aimed to conservatively design our machine psychophysics experiment, making the comparison between humans and machine as fair as possible, and the task as easy as possible for models to perform well on (See Section A.5). ***Finally, and most notably, we validate in a control experiment that human detection performance on TTM images closely follows performance on original images*** (Fig. 4). Thus, it is unlikely that DNNs were disadvantaged compared to humans in terms of the amount of information available.

## 8.2 EFFECT OF PERIPHERAL TRAINING

Training on the peripheral images in COCO-Periph reduces the gap in object detection performance. Interestingly, we see evidence that training helps more with object localization rather than identification (Figure 5) – this may explain why models trained on COCO-Periph do better on the psychophysics tasks than some models that have a higher AP on COCO-Periph. One role of peripheral vision is to guide where to fixate next, and favoring localization over identification aligns with the goal of guiding fixation. Despite these improvements, we observe that models trained on COCO-Periph still exhibit drops in performance that are greater than humans. This suggests that the behavior we see is not solely attributable to a domain shift. The long-term purpose of our dataset and experiments are to build new ways of matching human behavior beyond fine-tuning alone. Our results imply that task formulation is a critical area to explore in aligning DNNs and humans. TTM as a model suggests that one general representation can explain behavior on a variety of visual tasks. We believe an important future direction in bridging the gap between humans and DNNs is to optimize for generalization across a variety of tasks – rather than maximizing for accuracy on a single task. Current benchmarks in computer vision do not encourage this, and we hope that our dataset and experiments can facilitate research in this direction.

Training on COCO-Periph also increases robustness to geometric corruptions, but decreases robustness to noise corruptions. Although the texture-like representations of peripheral vision may contribute to human robustness to adversarial noise (Harrington & Deza, 2022), the TTM-transform itself more closely resembles geometric corruptions and this is evident in our robustness evaluations. While we do not evaluate the adversarial robustness of our trained models, it appears that more work is needed to fully understand the relationship between peripheral vision and robustness.

## 8.3 COCO-PERIPH – A NEW BENCHMARK WITH REAL-WORLD APPLICATIONS

A key contribution of our work is COCO-Periph, one of the first large scale datasets for studying peripheral vision in DNNs. We present an application of COCO-Periph in object detection, but COCO-Periph provides a unified and flexible framework for modeling peripheral vision in any modern DNN. By building the dataset on COCO, peripheral vision can be evaluated for the first time on tasks that go beyond simple crowding measures (Volokitin et al., 2017; Lonnqvist et al., 2020) and object/scene recognition (Deza & Konkle, 2020; Jonnalagadda et al., 2021; Pramod et al., 2022; Min et al., 2022). We present COCO-Periph, along with our psychophysics experiment, as a tool to the community to build DNNs that can predict human performance and benefit from properties of human vision.

Peripheral vision is useful to model and predict because plays a key role in guiding visual attention in humans, enabling fast, efficient detection of features and motion over a wide visual field. Peripheral representations in machine learning give performance boosts for object recognition (Pramod et al., 2022; Jonnalagadda et al., 2021; Min et al., 2022), and we extend this work to object detection. Modeling this enables us to better predict human visual behavior. This has applications in many areas including: (1) driver safety where we could predict if a person sees a hazard, (2) content memorability where we could optimize images to capture attention, (3) UI/UX to create displays that easy to view, and (4) foveated rendering, and (5) compression where peripheral vision could help models perform under reduced visual information.

## 9 ETHICS STATEMENT

Our study contributes to research modeling human visual behavior. Although we do not achieve human level performance, potential harms of such a system could include difficulty distinguishing AI from humans and tracking human attention patterns. Reducing the harm of both these risks could be achieved through watermarking or other signatures to make it clear when a model is being used. Regarding our human subject experiment, all participants provided informed consent prior to participation, in compliance with the Common Rule (45 CFR 46). This study was assessed as exempt from review by an Institutional Review Board, pursuant to 45 CFR 46.101(b)(2). There were no risks or discomforts associated with the study, beyond what is normally expected using a standard computer or video game. The eye tracker used was non-invasive and did not require special contact lenses. To minimize visual and motor fatigue, participants were given breaks every 15 minutes and were informed that they could exit the experiment at any point for any reason. All participant data was anonymized and kept in a locked room under password protection. See Appendix section A.3.2 for for further details.

### ACKNOWLEDGMENTS

This work was funded by the Toyota Research Institute, CSAIL MEnTorEd Opportunities in Research (METEOR) Fellowship, US National Science Foundation under grant number 1955219, and National Science Foundation Grant BCS-1826757 to PI Rosenholtz. The authors acknowledge the MIT SuperCloud Reuther et al. (2018) and Lincoln Laboratory Supercomputing Center for providing HPC resources that have contributed to the research results reported within this paper.

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

# A APPENDIX

## A.1 UNIFORM TTM METHODS

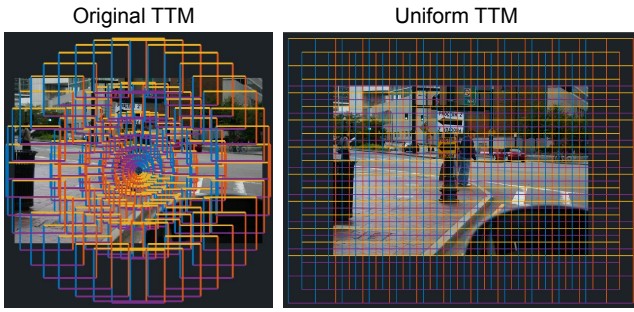

Figure 7: **Pooling Regions in Original and Uniform Texture Tiling Models (TTM).** Original TTM (Rosenholtz et al., 2012a) is foveated, so its pooling regions are small around the fixation point and grow farther from the fixation. We adapt TTM to use a fixed pooling region size everywhere in the image (Uniform TTM). The size is determined by the distance in the periphery being modeled.

In the original TTM model, the pooling region size is determined by a pooling rate, $r$, and a distance from the fovea, $d$. For the uniform version, we fix $d$ for a certain eccentricity rather than varying it like the original model. We set the overlap between pooling regions to be $60\%$, and we arrange the uniform pooling regions in a rhombic lattice to make it as close as possible to original TTM. We use the same synthesis procedure as original TTM (matching statistics like those defined in (Portilla & Simoncelli, 2000) for each pooling region iteratively from noise). The uniform TTM transforms take between $2-3$ hours to synthesize on $1$ CPU core (compared to the original TTM transforms, which take 6 hours on 1 core). Closer eccentricities like $5°$ take longer to run than large ones because the pooling region size is small. For COCO-Periph, we create uniform TTM transforms for $5,10,15$, or $20°$. For all TTM transforms, we assume that there are 16 pixels per degree, which is standard for original TTM.

When changing to uniform pooling, we also change the ordering of pooling region optimization from foveated TTM. Foveated TTM alternates spiraling from fovea to edge of periphery and back. This caused artifacts in uniformly-tiled TTM. Therefore, we opted for a randomly ordered optimization of the pooling regions, eliminating the optimization artifacts.

## A.2 PSEUDO-FOVEATION

Using the uniform TTM transforms, one can quickly simulate a fixation anywhere in an image, essentially generating a foveated TTM transform at low computational cost. This has been done previously for blur (Perry & Geisler, 2002; Geisler & Perry, 2002). We call the process pseudo-foveation. To create pseudo-foveated images, we stitch together uniform TTM transforms rendered at multiple eccentricities (Figure 8). For the fovea, we insert a circle crop of the original image. Then we add cropped rings from the uniform TTM transforms, centering each ring at the eccentricity it was rendered at ($5°$ degree TTM transform is centered at $5°$ eccentricity in the image). To reduce edge effects, we linearly blend the border between uniform TTM transform crops; all borders are weighted equally in the blending. With pre-computed transforms, this process of pseudo-foveation takes only 50ms per image, more efficient at generating images than foveated TTM, making it feasible to incorporate into the dataloading loop for training and testing DNNs.

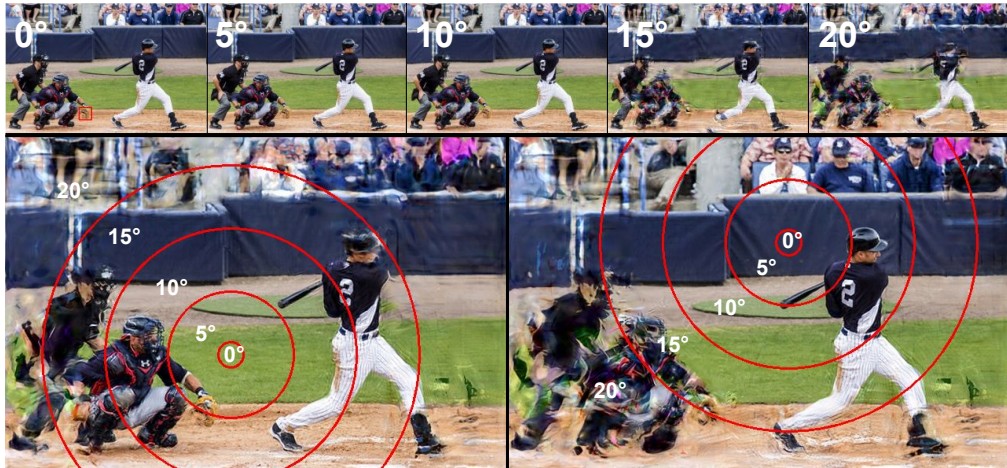

Figure 8: **Pseudo-Foveated images are created by stitching together Uniform TTM images.** Pseudo-foveation allows us to simulate a fixation at any point in an image. The top row shows TTM transforms for a single eccentricity, and the bottom row shows two pseudo-foveated TTM transforms for different fixation points.

### A.3 HUMAN PSYCHOPHYSICS EXPERIMENT

#### A.3.1 PRESENT / ABSENT EXPERIMENT IMAGE PAIRS

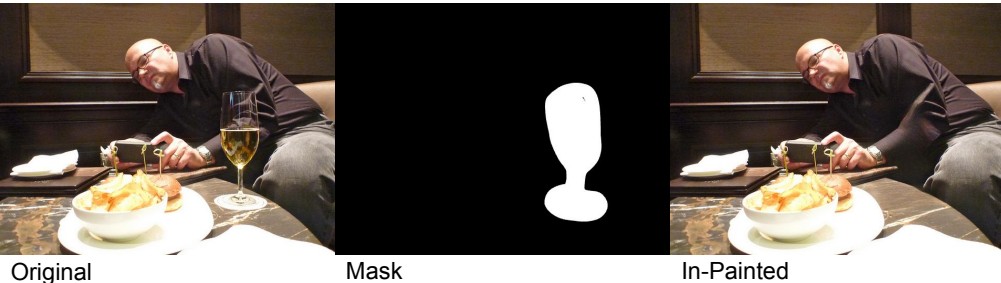

Original        Mask       In-Painted

Figure 9: **Example of LaMa (Suvorov et al., 2022) inpainting on COCO validation set image.** We use inpainting to create the object absent version of each image in the psychophysics experiment. Here, the wine-glass is removed with few artifacts.

To create pairs of images where a given object was both present and absent, we used images from the MS-COCO validation set (Lin et al., 2014) (such that they would be novel to both humans and to trained networks). We found images in landscape orientation where an object from a COCO object category appeared and was labeled only once in the image, and the object was detected with at least 50% confidence in the original image with the detectron2 (Wu et al., 2019) object detection model (faster_rcnn_r50_fpn). From this set, we hand-selected 26 images that spanned a range of conditions that would affect the difficulty of the peripheral detection task (object identity and size, variation in luminance and color contrast from background, amount of crowding around object, etc). We then used the LaMa image in-painting model to inpaint the chosen object (Suvorov et al., 2022), with a hand-drawn in-painting mask rather than the entire bounding box, as to avoid in-painting nearby objects in crowded scenes. In addition to the in-painted images, for each COCO image we also created a size matched $1/f$ pink noise mask to eliminate any motion transients and after-image effects during the experiment. These 26 image pairs were used for our object detection experiments. Note that figures reflect 24 images, as 2 images were removed from analysis because of poor psychometric curves fits (see Figures 20 and 21 to view the final image set).

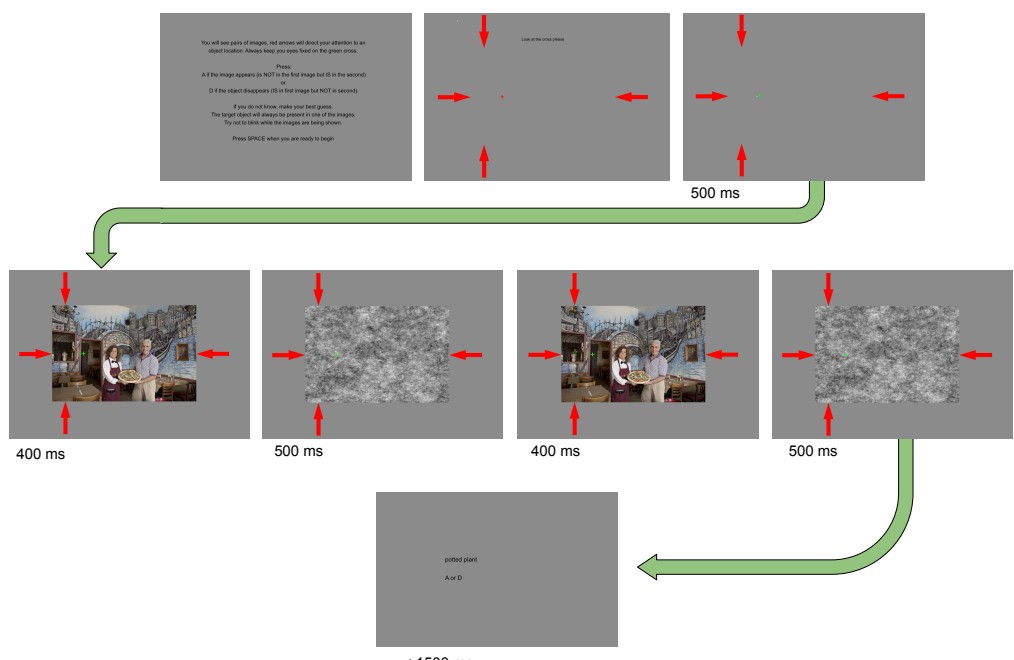

Figure 10: Human psychophysics experiment trial. Subjects complete a 2IFC (2 interval forced choice) task where they determine if a target object appears or disappears in a sequence. Red arrows indicate the location of the target object. Subjects fixate at a cross that is placed at $5, 10, 15$ or $20°$ away from the object.

### A.3.2 EXPERIMENTAL SETUP

All participants provided informed consent prior to participation, in compliance with the Common Rule (45 CFR 46), and this study was assessed as exempt from review by MIT's Institutional Review Board, pursuant to 45 CFR 46.101(b)(2). Participants took approximately 2 hours to complete the study and were paid a $40 Amazon gift card for their participation.

12 subjects participated in the human psychophysics experiment. We discarded the data from 2 subjects due to a computer malfunction and difficulty eye-tracking with a strong contact lens prescription. The remaining subjects consisted of 4 Male, 5 Female, and 1 Non-binary subjects ranging in age from 19 to 31. All had self-reported normal or corrected to normal visual acuity with contact lenses, with no history of eye surgery. 2 subjects had corrective lenses for myopia with a correction less than -1.25, but did not normally wear glasses or contacts (and did not during the experiment); We included these subjects as the viewing distance was only 82cm.

Subjects were seated and head placed in the chinrest of an EyeLink 1000 in tower-mount configuration. Subjects were 82 cm from a monitor screen, and their left eye position tracked. Nine-point calibration was performed and validated to within 1 degree at each point. The experiment allowed for fixation within 2 degrees, displaying a small dot on the screen for real-time feedback of measured fixation location. Subjects were asked to pause the trial block to re-calibrate if measured fixation did not reflect fixation location, or they had difficulty with the system recognizing their fixation.

### A.3.3 EXPERIMENTAL PARADIGM

The experiment consisted of a 2IFC (two interval forced choice) task where subjects report which out of two images contains a target object. Each subject saw 26 image pairs 10 times at 4 different fixation locations $(5, 10, 15, 20°)$ away from a target object, where the fixation location was at the vector computed from the target object location towards the center-point of the image. The order of each presentation was randomized across the whole experiment. Each subject saw the present/absent

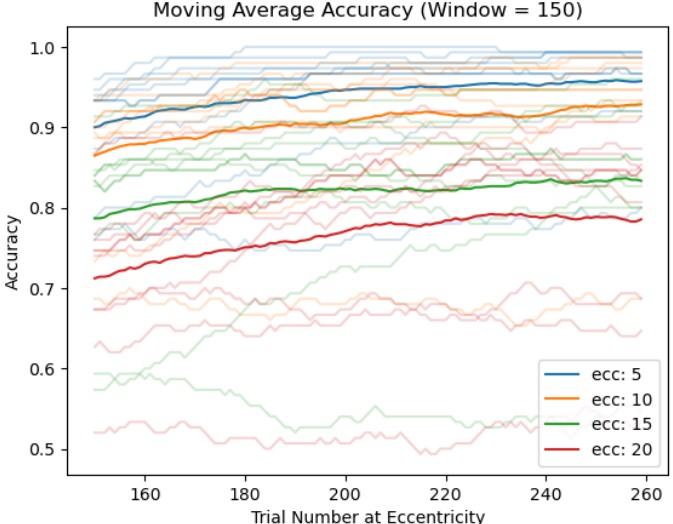

Figure 11: **Learning Over Experiment.** Despite including practice trials before the experiment and excluding correct/incorrect feedback during the experiment, human responses did exhibit a small learning effect over the course of the experiment. Accuracy is plotted against x-axis ordered by the number of times subject had seen any image at a given eccentricity. Bold lines show moving window averaged accuracy over all subjects, and pale lines show individual subject's data.

image pair 5 times present-first and 5 times absent-first. No correct/incorrect feedback was given to the subject.

Subjects maintained fixation on a cross presented at either 5, 10, 15, or 20 degrees from the object location, and were eye-tracked to ensure fixation was maintained within 2 degrees. Attention was directed to the object location with latitude/longitude arrows. After presentation, subjects were given the original COCO object category, and prompted to report which image contained the the object by reporting if the object 'appeared' (was in the 2nd image but not the first) or 'disappeared' (was in the 1st image but not the second).

Each trial waited to begin until the subject fixated on a cross before proceeding. If the subject broke fixation anytime an image or mask is shown, the trial was aborted and shuffled to the end of experiment. Each image was shown for 400ms, and a size-matched pink noise mask was shown after for 500ms to eliminate visible flicker of appearance of disappearance. Subjects were given a response window of 1.5 seconds, and the image pair shuffled to the end of the experiment in a time-out.

The experiment in total was 1040 trials long. Subjects were given a break every 150 trials, recalibrating after each break and before starting a new block. Before beginning the experiment, subjects completed a practice round consisting of 15 trials of very easy image pairs. 2 subjects needed to do the practice round a second time before they reported being comfortable with the task. The images in the practice round were much larger than those in the actual experiment to make the task easier (10+ degrees). This may have contributed to a learning effect we observed in some subjects where performance improves with the number of trials completed (Figure 11).

## A.4 HUMAN CONTROL EXPERIMENT: MEASURING HUMAN PERFORMANCE ON UNIFORM TTM IMAGES

To rule out the hypothesis that DNN under-performance could be due to TTM throwing away too much information, we performed a control experiment measuring human detection performance on uniform TTM images. Again, all participants were provided informed consent prior to participation. This experiment consisted of 3 naive subjects and 2 expert subjects, 3 males, 2 females.

In this experiment, we used the same 2IFC task as the original image experiment. Instead of asking participants to fixate a set distance away from a target object, participants were able to freely view a Uniform TTM images simulating $5, 10, 15, 20°$ viewing conditions. Images were presented in a 22 pixels per degree set-up, similar to the original experiment. To avoid a search task, we cued the location of the target object before each trial. Stimulus, mask, and response timings were the same as the original experiment (400ms, 500ms, 1.5s respectively). Like the original, the experiment was 1040 trials total, consisting of 26 image pairings shown for 4 eccentricities for 10 repetitions. 5 of the repetitions showed object present images first and the other 5 showed object absent first. For these repetitions, we used 5 unique seeds of Uniform TTM images. Participants were given breaks every 150 trials.

### A.4.1 HUMAN PSYCHOPHYSICS RESULTS

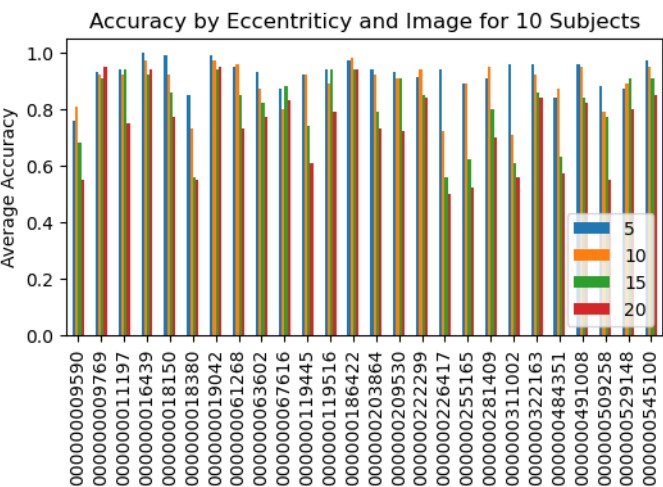

Figure 12: **Per-Image Accuracy over all Subjects.** When viewing original images at varying eccentricities, human performance was very image-dependent, but decayed as eccentricity increased for all images

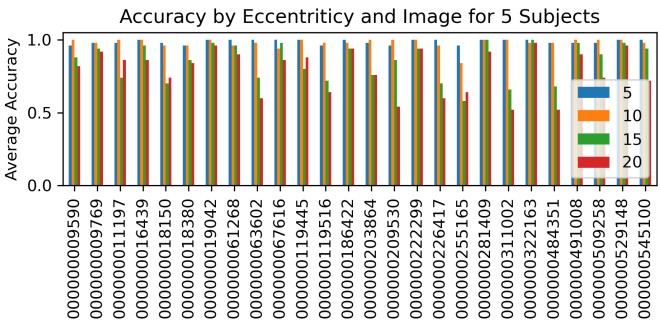

Figure 13: **Per-Image Accuracy for UniformTTM images over all Subjects.** When viewing uniform TTM images, human performance was also image-dependent, and also decayed as eccentricity increased for all images.

Overall, subjects performed well at the task for most images, only reaching chance performance for approximately 30% of the images (Figure 12). Images 000000009769 and 000000067616 were removed from downstream analysis due to poor fitting of psychometric function. The difficulty was extremely image-dependent - subjects reported similar feedback during debriefing that certain images were extremely difficult and that they were guessing (though results show they often performed better than chance, despite this), while other images were extremely easy. The hardest and

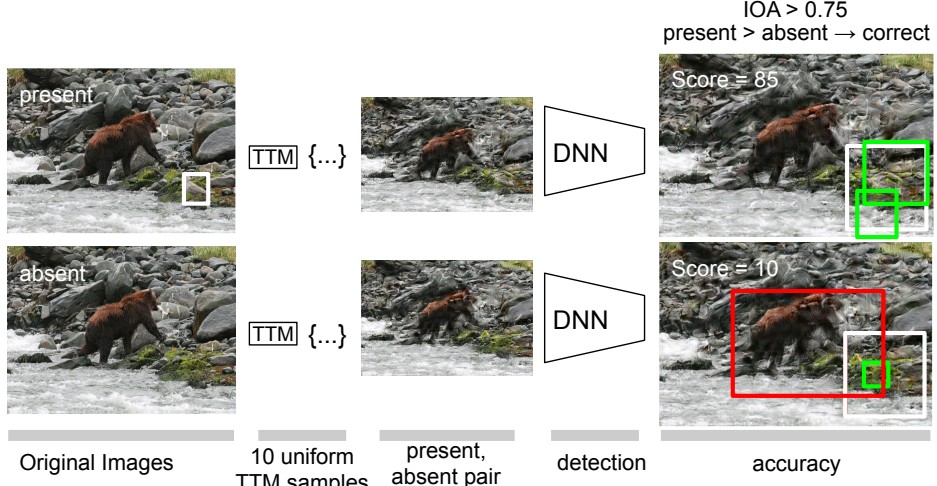

Figure 14: **Workflow for Machine Psychophysics Experiment.** To simulate trials for detection DNNs, we generate 10 uniform TTM transform images for each present/absent image in the experiment. We run inference on the transforms and sum the scores of all box predictions that have an intersection over area (IOA) greater than $0.75\%$. If the summed score is greater on the present transform, the DNN is recorded as correct for that pairing. White boxes indicate the ground truth and padded ground truth. Green boxes show predictions that meet the area condition, red do not.

easiest reported images tended to be those least and most crowded, and those with the most and least background contrast, respectively. We see similar results for subjects viewing TTM images directly (Figure 13).

## A.5  MACHINE PSYCHOPHYSICS EXPERIMENT

We provide pseudo-code for the machine psychophysics procedure. We tried a variety of detection criteria including, enforcing that predictions match the target category, enforcing the size the box predictions to be no more than half or twice the size of the padded ground truth, and taking the average score over all boxes that overlap the padded ground truth. We arrived at the summing approach described in the Algorithm1 because it yielded the highest critical $\mu$ scores and showed similar trends in performance to the other approaches. For some models, there were 1-2 images for which the psychometric fit did not converge, and therefore a value for $\mu$ could not be determined.

In addition, a potential edge case in our DNN psychophysics setup is when the present image probability equals the absent image probability. We investigated how often this happened and found that for the baseline R50-RCNN model, this represents only cases where the prediction probability is zero (no predictions above the 1% threshold) in both the present and absent mongrel pair. Over the image set, it occurs at least once over the 100 present/absent pairings, in 10/26 images, and for each eccentricity ($80°,160°,240°,320°$), occurs (70,286,357,239) times over the 2600 pairings per eccentricity, for a total of (2.7%,11%,13.7%,9.2%) overall. For the baseline RCNN, we observe that aprob never equals pprob because the model had equivalent predictions in the present and absent images. This only occurs when no objects are predicted in either image.

Making an apples to apples comparison between humans and machines, especially for object detection, is non-trivial; in formulating a psychophysics experiment for these models, which are trained to perform very specific tasks, and report only specific information, design choices inherently affect performance. We took a conservative approach, aiming to give models as much of an chance as possible at performing on-par to their human counterparts. We note here various ways in which we allowed models flexibility in the machine psychophysics experiment:

1. For the experiment, we chose images with present/absent objects for which models were able to detect confidently for the original ($0°$) image.

**Algorithm 1** For each object present/absent image in the human experiment, we create 100 pairings of uniform TTM transform images ($P$ and $A$). We simulate trials by looking at the box predictions ($boxes$) of a detection DNN for each pairing ($p, a$). We sum the total box scores that overlap with the target object box at least $0.75\%$ IOA (intersection over area). To determine this overlap, we take the target object ground truth box ($gt$) and pad it with by half a pooling region ($pr$). If the total score for the present image ($pprob$) is higher than the absent ($aprob$), we record the DNN model as having a correct response. We average over all 100 pairings for final accuracy ($acc$).

---

**procedure** GETMODELACCURACY
 $acc = 0$                    ▷ initialize accuracy
 **for** $p, a \in (P, A)$ **do**      ▷ loop through all present/absent pairings for one object
  $pprob = \text{GetTargetDetectionScore}(p, gt, pr)$     ▷ get score for target object
  $aprob = \text{GetTargetDetectionScore}(a, gt, pr)$
  **if** $pprob = aprob$ **then** $acc = acc + 0.5$       ▷ get per trial accuracy
  **if** $pprob > aprob$ **then** $acc = acc + 1$
 $acc = acc \div trials$            ▷ take average over all trials
**function** GETTARGETDETECTIONSCORE($im, gt, pr$)
 $gtx = gt + 0.5 \times \text{size}(pr)$    ▷ expand ground truth bounding box by half pooling region
 $boxes, scores = \text{DNN}(im)$           ▷ get box proposals
 $prob = 0$         ▷ initialize total score of overlapping proposals
 **for** $b, s \in (boxes, scores)$ **do**
  **if** $\text{ioa}(gtx, b) > 0.75$ **then**     ▷ check boxes that overlap expanded ground truth
   $prob = prob + s$
 **return** $prob$           ▷ return sum of overlapping scores

---

2. We utilized a padded bounding box when evaluating predictions, as TTM has location invariance on the order of a pooling region width that can move objects in transformed images. To compensate for this, we included any predictions with an IOA of ¿0.75. In addition, human subjects were primed with an object location, but were not restricted within a specific region of the image to attend when making the present/absent determination.

3. We did not enforce models to match labels when performing the present/absent task, allowing thee prediction of any object to be counted. Humans may have been able to perform the present/absent task by simply reporting the present image as the one with more 'objectness', without regard to label. We also found removing the label requirement improved model performance.

4. We measured total probability as the sum of all prediction probabilities that overlapped the bounding box, rather than the maximum. This was chosen for the same reason as (1), in that humans could have used a similar metric. In addition, we found that summed probabilities showed better model performance as opposed to maximum probabilities (Figure 27).

## A.6 CONTROL EXPERIMENTS: ORIGINAL TTM, PSUEDO-FOVEATED, AND CORRELATION WITH HUMAN PERFORMANCE

For our main results, we report DNN performance using uniform TTM images. As a control we also run machine psychophysics performance on original TTM (Figures 15) and pseudo-foveated (Figure 16). We see similar trends in performance to Figure 4 of the main text. We also look at the correlation between the human and DNN performance across each image set in Figures (17, 18, 19).

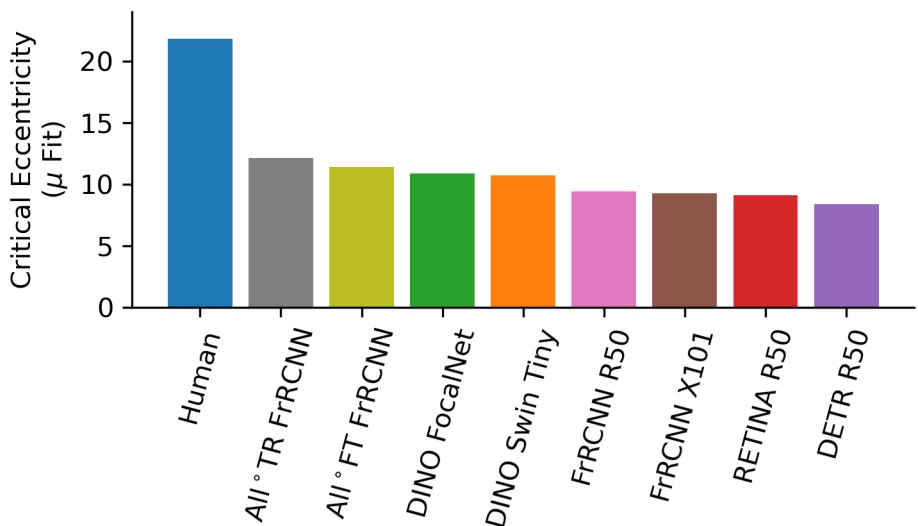

Figure 15: **Critical Eccentricity using original TTM in the machine psychophysics**.

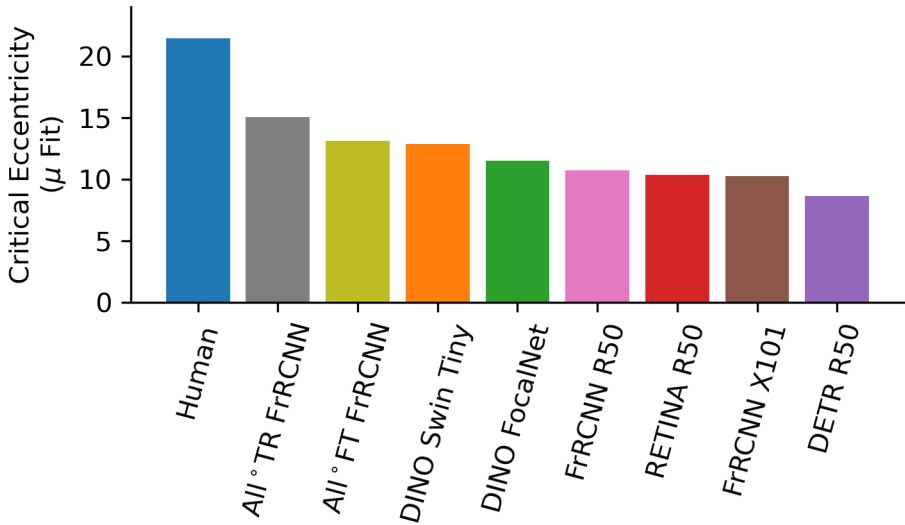

Figure 16: **Critical Eccentricity using pseudo-foveated TTM in the machine psychophysics**.

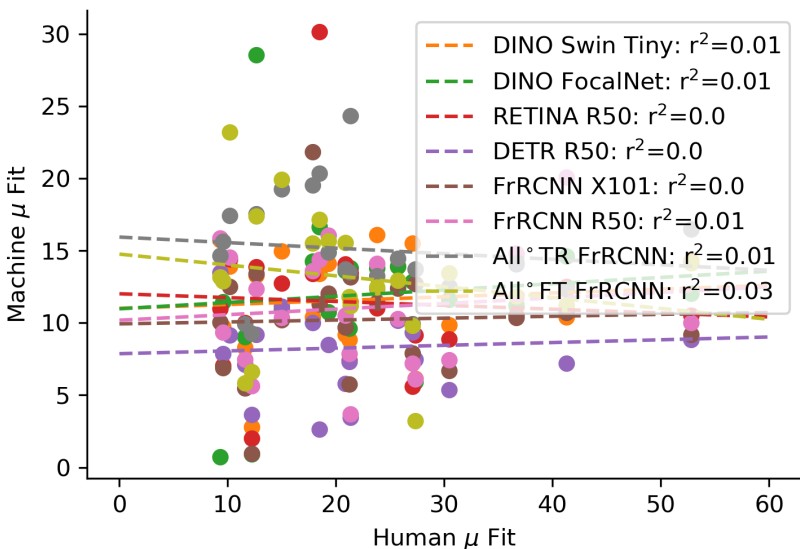

Figure 17: **Correlation between Human and Machine Critical Eccentricity using uniform TTM in the machine psychophysics.**

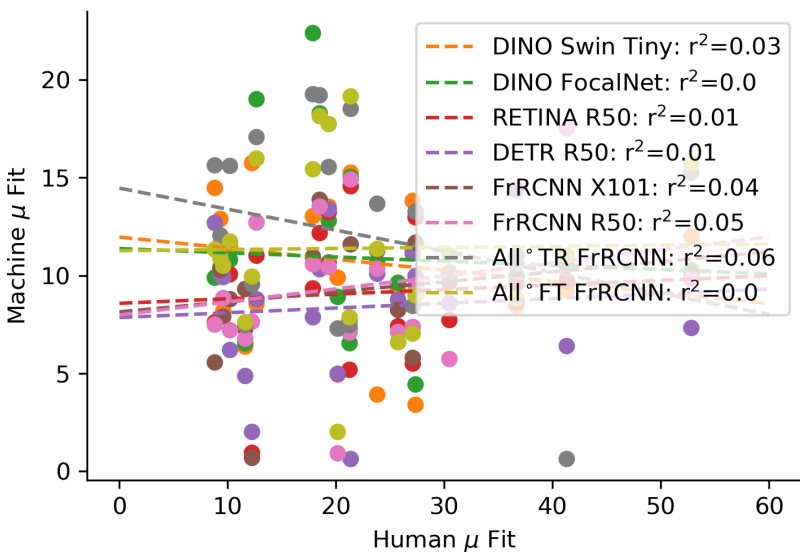

Figure 18: **Correlation between Human and Machine Critical Eccentricity using original TTM in the machine psychophysics.**

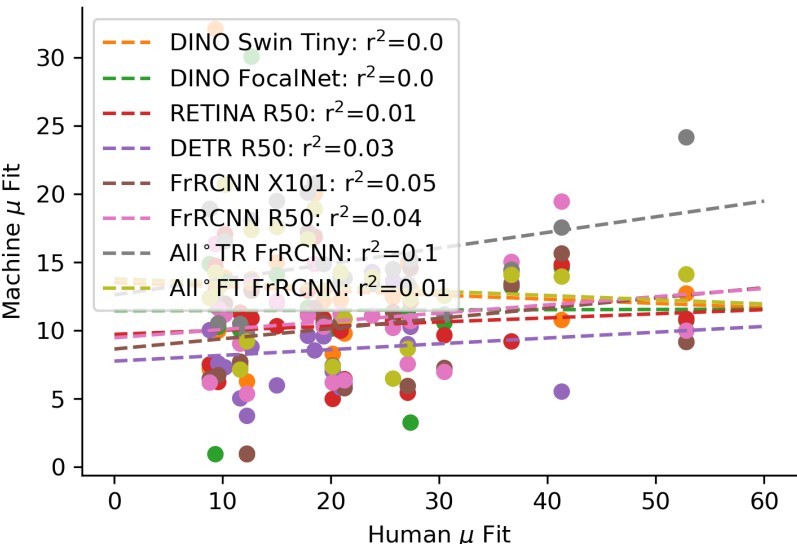

Figure 19: **Correlation between Human and Machine Critical Eccentricity using pseudo-foveated TTM in the machine psychophysics.**

# Present Experiment Images

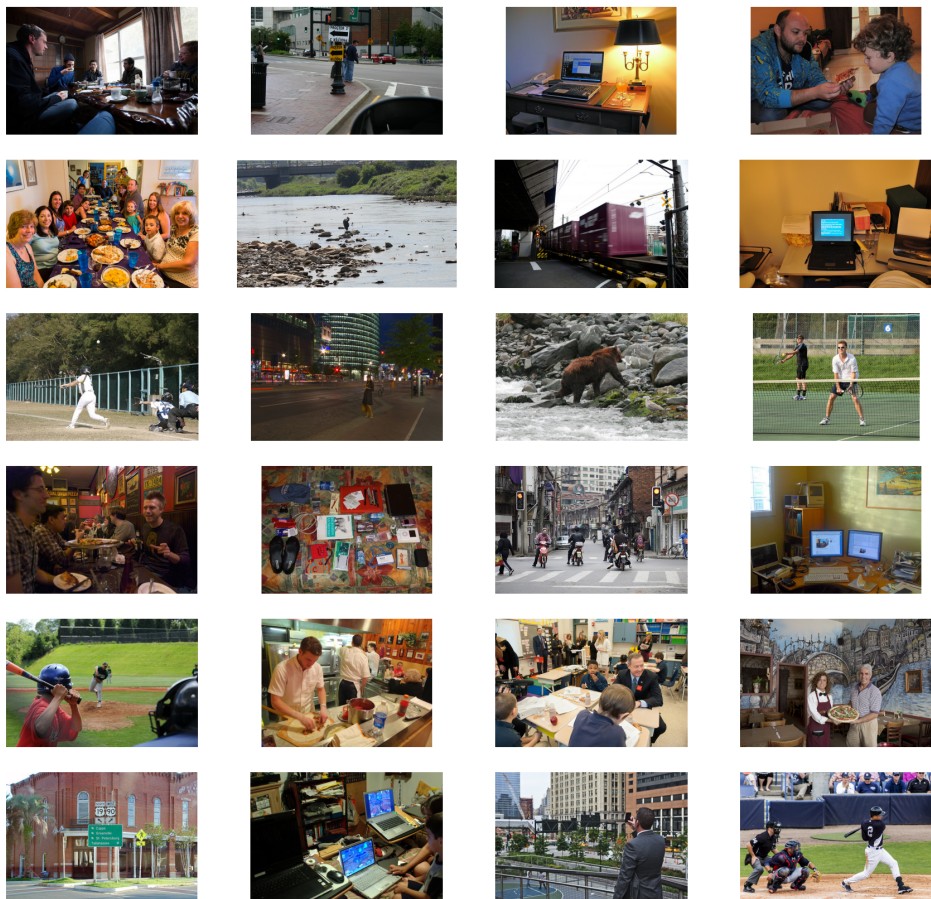

Figure 20: **Object Present Images.** We selected 24 images from the original COCO validation set to use in a psychophysics experiment measuring human object detection in the periphery.

## Absent Experiment Images

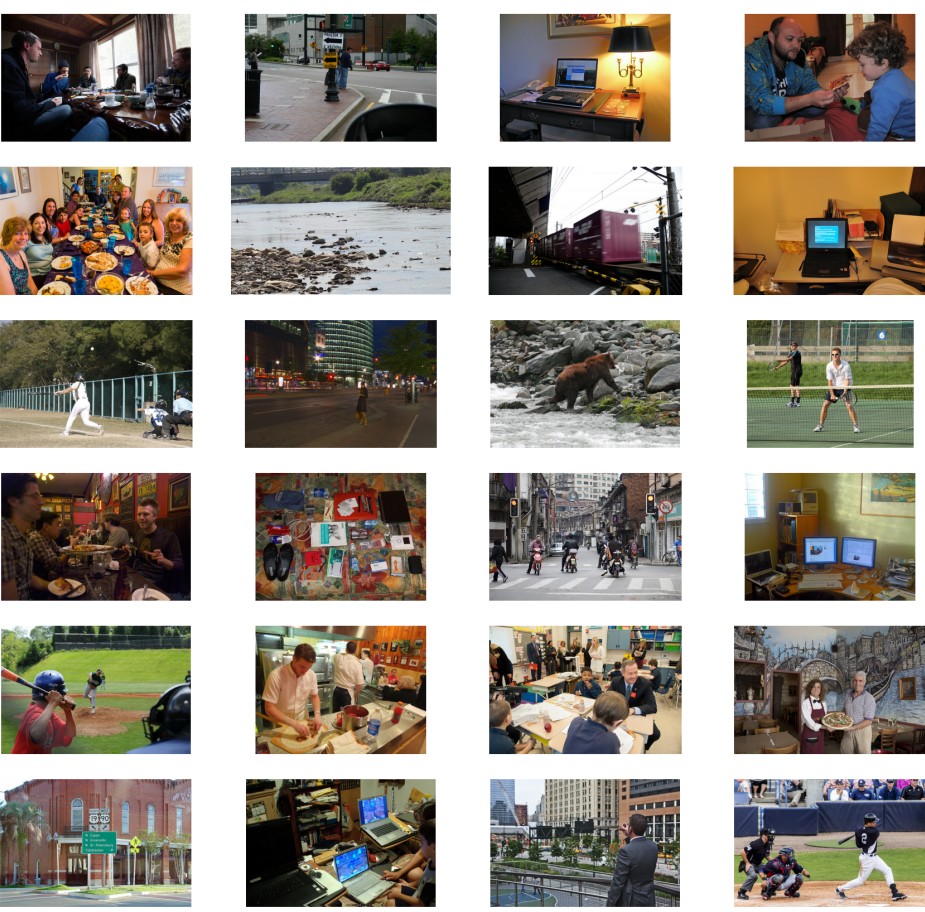

Figure 21: **Object Absent Images.** We in-painted out a target object from 24 original COCO validation set images to use in the human object detection experiment.

### A.7 FINE-TUNING OBJECT DETECTION MODELS

### A.8 TRAINING PROCEDURE

| Model | Img | 5° UTTM | 10° UTTM | 15° UTTM | 20° UTTM |
|---|---|---|---|---|---|
| $0°FT$ Ecc | 100% | - | - | - | - |
| $5°FT$ Ecc | 50% | 50% | - | - | - |
| $10°FT$ Ecc | 50% | - | 50% | - | - |
| $15°FT$ Ecc | 50% | - | - | 50% | - |
| $20°FT$ Ecc | 50% | - | - | - | 50% |
| All$°FT$ Ecc | 20% | 20% | 20% | 20% | 20% |
| All$°TR$ Ecc | 20% | 20% | 20% | 20% | 20% |

Table 3: Distribution of Original COCO training set images (Img) and varying eccentricity TTM transform images (UTTM) used for fine-tuned Object detection models. Transform images are derived from COCO training set images.

We fine-tuned the Faster R-CNN model from the Detectron2 library (Wu et al., 2019) (faster_rcnn_r50_fpn) using a mixture of original training images, and TTM transforms for varying

eccentricities. In training, we selected 55,000 images from each of the COCO and COCO-Periph training sets at each of the 4 eccentricities (5,10,15,20). We train an additional model on all eccentricities using 55,000 images for each eccentricity (see Table 3 for image distributions; the model fine-tuned on $5°$, for example, is trained on 55,000 COCO images and 55,000 COCO-Periph $5°$ transform images). All models are trained for 180,000 iterations starting from the weights of a pre-trained R-CNN from (Wu et al., 2019). We set the solver to step at 120,000 and 160,000. We set the base learning rate to $3 \times 10^{-4}$. All other training parameters are the same R-CNN training parameters in (Wu et al., 2019) as the baseline model.

We also train one model from scratch on all eccentricities (55,000 images from each eccentricity as well as 55,000 original COCO images). We use the same 3x training schedule provided in (Wu et al., 2019) for Faster RCNN R50 FPN models (starting from an ImageNet trained ResNet50 backbone, training for 270,000 iters, 16 images per batch). Due to computational limiations, we do not fine-tune or train any transformer-based object detection models; we only analyze their baseline models which we obtain from the detrex library (detrex contributors, 2022). In the main text, we report results for fine-tuning on all eccentricities since that model is the top performer.

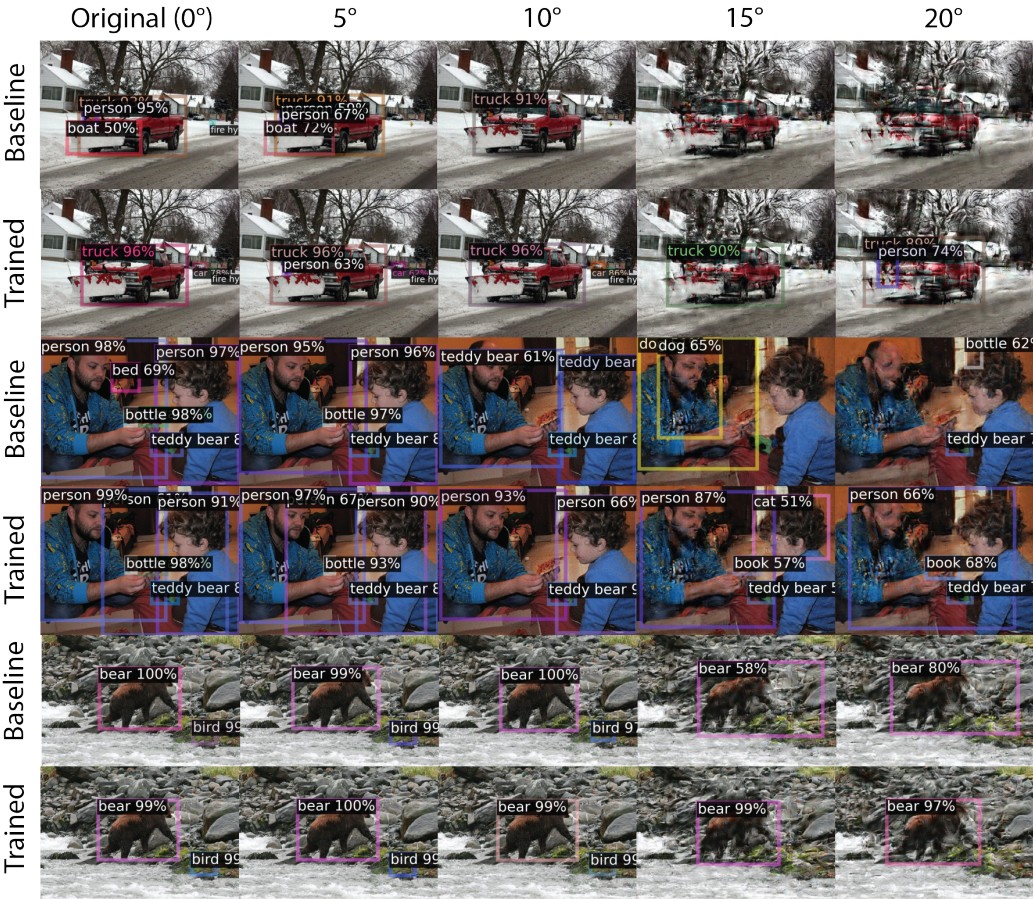

Figure 22: **Training improves object detection on Uniform TTM images.** Faster-RCNN R50 FPN trained from scratch on peripherally transformed images retains more stable predictions with more accurate bounding boxes as eccentricity increases when compared to the baseline Faster-RCNN R50 FPN model.

### A.8.1 AVERAGE PRECISION OVER ALL, SMALL, & LARGE BOUNDING BOXES

We evaluate all fine-tuned models on the COCO-Periph validation set and compare detection performance for small and large bounding box objects. We find that fine-tuning on uniform TTM images improves performance on the COCO-Periph validation set more for large objects than small objects.

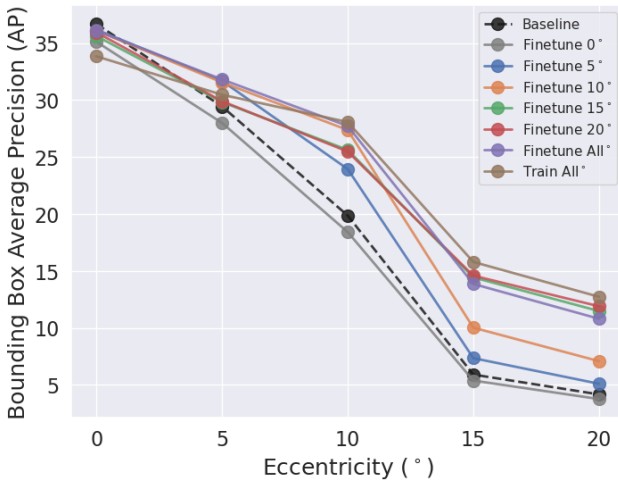

Figure 23: **AP on COCO and COCO-Periph validation for COCO-Periph Trained Models.** We train and fine-tune Faster-RCNN-R50-FPN model on TTM transformed images from the COCO-Periph train set, and show improvement in AP score on COCO-Periph validation, with minimal reduction in performance on original COCO validation set. Finetune 0° is trained on original COCO training set, Train All° is trained from only the ResNet-50 backbone on COCO-Periph training set images.

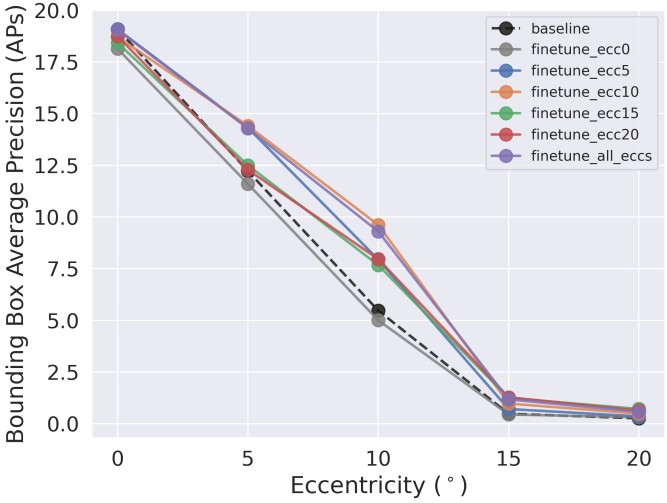

Figure 24: **Faster R-CNN R50 Object Detection Bounding Box AP for Small Objects.** We fine-tune an R-CNN on TTM transform images and compare their AP small to the baseline model. All models generally do poorly on smaller objects. Fine-tuning does not improve performance on original COCO images, but it does slightly improve performance on small eccentricity COCO-Periph images like 5 and 10°.

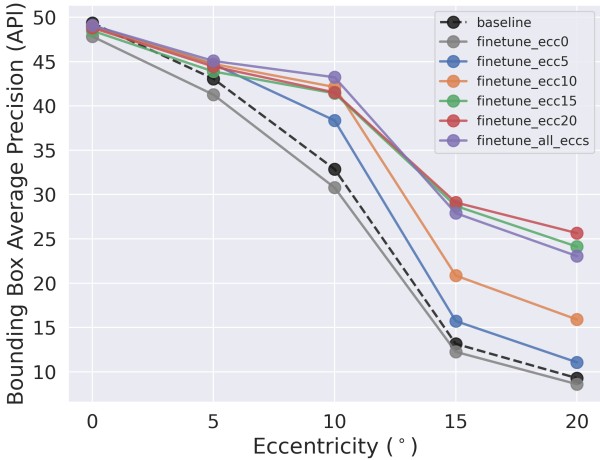

Figure 25: **Faster R-CNN R50 Object Detection Bounding Box AP for Large Objects.** All models generally do better at large objects than small objects, and fine-tuning on uniform transform images improves large object performance on COCO-Periph images much more than small objects at farther eccentricities.

## A.9 Robustness to Corruption in Fine-Tuned & Trained Faster R-CNN

Fine-tuning and training from scratch on uniform TTM transform images improves robustness to some unseen geometric corruptions, but not contrast and noise-like corruptions (Figure 26).

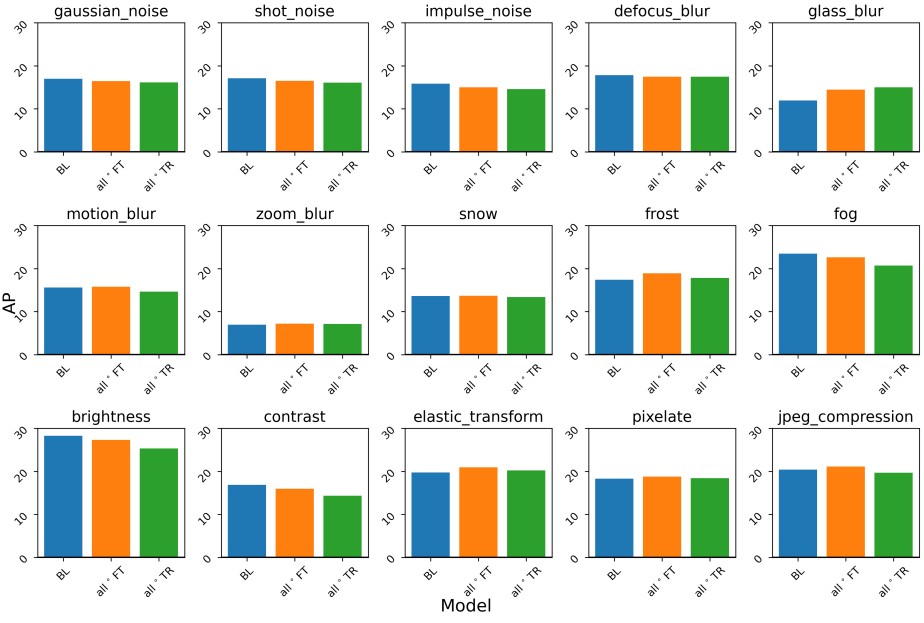

Figure 26: **Corruption Robustness of Fine-Tuned and Trained models.** We report AP averaged over all 5 severity levels for each corruption in COCO-C (Michaelis et al., 2019). BL refers to the pre-trained Faster RCNN R50 model (blue). The FT model is one fine-tined on COCO-Periph (orange). The TR model is trained from scratch on COCO-Periph (green).

## A.10 MAXIMUM PROBABILITY AS PREDICTION REDUCES PERFORMANCE

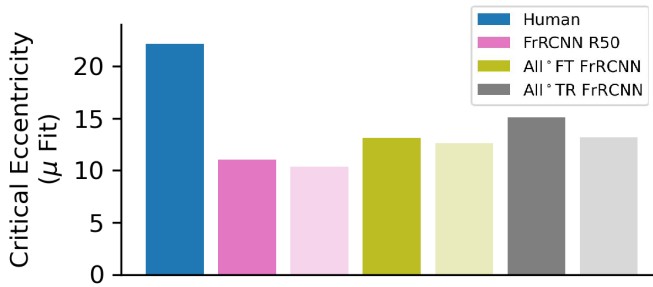

Figure 27: **Sum of Probabilities vs Maximum Probability** Using the sum of the prediction probabilities (opaque) as the prediction results in improved performance on the 2IFC task as compared to using the maximum predicted probability (translucent). This holds for both the baseline, fine-tuned, and trained RCNN models.

