# OpenReview forum: "COCO-Periph: Bridging the Gap Between Human and Machine Perception in the Periphery"
_ICLR.cc/2024/Conference — ICLR 2024 poster_

### Official Review · Reviewer_eW1M · 2023-10-31

**Soundness:** 2 fair
**Presentation:** 3 good
**Contribution:** 1 poor
**Rating:** 3
**Confidence:** 4

**Summary:**

The paper explores the performance of deep neural networks when constrained by human peripheral vision, using a modified TTM model and proposes a new dataset called COCO-Periph. A psychophysics experiment with humans is conducted, aiming to highlight difference between human and artificial models in periphery-sensible tasks.

**Strengths:**

- The paper aims at investigating a very relevant topic, somehow underexplored in the past, when models of foveated vision normally only take into account loss of acuity.
- The paper introduces an efficient variation of TTM.

**Weaknesses:**

- The paper introduces a new dataset called COCO-Periph. However, it is unclear how this dataset significantly contributes to the existing knowledge. Given that the primary focus is on modeling peripheral vision in DNNs, the dataset itself does not seem to offer substantial value to the research community. A better approach would be to release the code for the community to experiment with, rather than creating a dataset that doesn't seem to add much to the field.
- If we interpret peripheral image transformation via TTM as a form of image perturbation, results are unsurprising. The observation that models perform poorly on shifted domains (in this case, TTM generated images) and become more robust after training on the new domain (COCO-Periph) is not surprising. The results are, in essence, not only predictable but do not provide an insight on periphearl vision. To be clear: the exact same results could have be obtain with ANY image transformation resulting in a domain shift.
- The paper fails to provide a clear explanation for why the proposed model, even after fine-tuning on the entire dataset, still performs significantly worse than DINO. This raises questions about the efficacy of the proposed model and whether it addresses the problem of peripheral vision adequately.
- Figure 5, which assesses the model's resilience, is presented misleadingly. It only demonstrates the model's performance across various eccentricities, which is expected, given that the model was trained on the same data. Still, as we see from Table 1, the proposed finetuned approach performs worse than DINO. It would be interesting to see absolute performance on 20 degs, for DINO and the proposed model (I imagine, that would not show any advantage)
- The paper includes an experiment comparing humans presented with objects at different eccentricities with a model presented with a uniform peripheral transformation obtained with TTM. This comparison is not fair. This essentially tests the model on out-of-distribution data, making the poor performance of the models obvious. In no way I see a support for any conclusions regarding properties of the peripheral vision.
- The paper lacks a clear explanation of the practical importance of a peripheral vision model. The only advantage seems to be an increase in robustness for very related image transformations...

**Questions:**

- Could you clarify the specific contributions of the COCO-Periph dataset, as it appears to have limited value in the context of the study? - The results in Table 1 show that the fine-tuned model still underperforms compared to DINO. Can you provide more insights into why this performance gap exists and whether there were any unexpected challenges in fine-tuning?
- Could you explain the reasoning behind fig. 5 and why a direct comparison of model performance with DINO in absolute values was not included?
- Can you provide more context on potential real-world applications or implications of this results?

**Details Of Ethics Concerns:**

any study aiming at modeling human behavior should include an analysis of potential harms.

---

> ### Author Response · Authors · 2023-11-19
> **response to review (part 1)**
>
> We thank reviewer eW1M for their thoughtful comments and questions.  Please see our response to your concerns as follows:
>
> > "the dataset itself does not seem to offer substantial value to the research community"
>
> We agree that code would be useful for further experimentation and plan to release it alongside the dataset. That said, we believe a dataset is also a valuable contribution because the peripheral vision model we would like to make available to the community takes ~5-6 hours per image to render. The entire dataset took over $6$ months to create. The procedure also depends on choosing a fixation point beforehand. To render multiple fixations at large scale during training is thus intractable. Furthermore, by providing a dataset, we are standardizing evaluation for future work. If we share only code, everyone will generate slightly different versions of the dataset, making it difficult to evaluate progress.  We have made these points more clear in the introduction.
>
> > "If we interpret peripheral image transformation via TTM as a form of image perturbation, results are unsurprising"
>
> We appreciate the reviewer eW1M’s insight about domain shift, but as noted by other reviewers, our results are interesting in the broader context of human vision. When humans view TTM images, their performance is highly correlated with original images -- despite never having “trained” on this exact image perturbation. If a DNN learned the same representational properties as human vision during training, then arguably they too should be able to handle this transformation. Yet, we observe that DNNs do not perform well under this transformation. As an experiment, we explore simply fine-tuning and training models on COCO-Periph to understand this behavior. We find that these models still exhibit drops in performance greater than humans do, suggesting the behavior is not just about domain shift. In the future, we ideally will have new models that do not require extensive fine-tuning -- just as humans do not require this. With our dataset and human experiment, our long-term goal is to build new methods that do perform like humans. We see COCO-Periph as a stepping-stone toward that goal, informing how to build representations robust to a peripheral transformation. We have added some of these points in the Discussion.
>
> > "The paper fails to provide a clear explanation for why the proposed model, even after fine-tuning on the entire dataset, still performs significantly worse than DINO"
> >"Figure 5, which assesses the model's resilience, is presented misleadingly. It only demonstrates the model's performance across various eccentricities, which is expected, given that the model was trained on the same data ..."
>
> First, we would like to emphasize that our main contribution is a dataset and human experiment, not a new proposed model. We include a fine-tuning experiment as an additional evaluation, but our larger goal is to uncover methods that do not require extensive optimization on TTM to perform well on this OOD transformation like humans do.
>
> We appreciate reviewer eW1M’s request for per eccentricity psychophysics performance in Figure 5. We agree that this is useful information, and we have added a subplot of these values in Figure 5. We originally chose to report an across eccentricity value ($\mu$) because it is standard practice in psychophysics studies to fit a curve across stimulus conditions and report a critical value where performance deteriorates to 75% accuracy (Strasburger, 2020, Figure 12), and we aimed to make an apples to apples comparison between humans and models.
>
> Regarding the results themselves, we agree it is puzzling that the fine-tuned model out-performs DINO when it under-performs on AP. We believe this can be explained by the nature of the TTM transform and psychophysics task we propose. By design, TTM perturbs the true object location -- especially for far eccentricities. When we report AP, we do not account for this and rely on the original COCO bounding boxes. We acknowledge that this is a limitation, and so we do account for it in the psychophysics task by padding the ground truth bounding boxes (see Section 6 and Appendix A.4).
>
> In addition, the psychophysics task itself only accounts for object localization and does not account for recognition performance like AP does. To test if this played a role in our results, we evaluated recognition alone for the RCNN in Figure 6b. Here, we see that the fine-tuning has a less noticeable improvement in performance; training from scratch even reduces performance. This indicates that object localization, rather than recognition, drives the performance boost we see in Figure 5. Indeed, peripheral vision is thought to primarily play a role in guiding where to look next rather than in recognizing object identity. For this reason, we evaluate models against humans on this metric and expect to see more improvements in this area when mimicking peripheral vision.

---

> > ### Author Response · Authors · 2023-11-19
> > **response part 2**
> >
> > > "The paper includes an experiment comparing humans presented with objects at different eccentricities with a model presented with a uniform peripheral transformation obtained with TTM. This comparison is not fair. This essentially tests the model on out-of-distribution data, making the poor performance of the models obvious. In no way I see a support for any conclusions regarding properties of the peripheral vision."
> >
> > Based on our results and prior experiments with TTM, we again propose that it is fair to test models on TTM transformed images because humans can effectively perform tasks with them despite never having trained on them. Studying DNN performance on such OOD data is thus critical, and we show that DNNs struggle in this regard. Prior work has established for multiple tasks including object detection that human performance on original and TTM transformations strongly match one another (Ehinger & Rosenholtz). This begs the question, what features do humans rely on to perform visual tasks in their periphery? Should these be learned or hard-coded in DNNs? By showing DNNs these transformed images and comparing performance to humans, our experiment helps the community answer these questions.
> >
> > > "The paper lacks a clear explanation of the practical importance of a peripheral vision model. The only advantage seems to be an increase in robustness for very related image transformations…"
> >
> > We thank reviewer eW1M’s for requesting a more clear explanation for why peripheral vision is useful to model. We acknowledge that our results do not demonstrate a concrete performance boost on standard benchmarks. However, we argue that peripheral vision is useful to model because peripheral vision plays a key role in guiding visual attention in humans, enabling fast, efficient detection of features and motion over a wide visual field By modeling peripheral vision, we can leverage the human visual system as a feature selector. This has already been shown to improve performance in object recognition (Pramod 2022, Jonnalagadda, 2021, Min 2022) . Peripheral vision, however, has not been modeled in natural scene detection tasks where it arguably plays a much bigger role than recognition. We agree that we do not find a peripheral vision transformation alone to be sufficient to improve baseline original COCO performance. We believe this is an interesting contribution suggesting modeling peripheral vision is more beneficial for some tasks than others. We have included some of these points in the discussion.
> >
> > > "Could you clarify the specific contributions of the COCO-Periph dataset"
> >
> > Please refer to our response to weakness #1-3.
> > * In summary, TTM is extremely computationally expensive, so the dataset makes it tractable to train on the transformation and explore some of the domain-shift questions reviewer eW1M aptly points out.
> > * Fine-tuning under-performs on AP likely inherent localization shifts introduced by TTM. In addition, our fine-tuned model was only trained on a subset of the COCO-Periph dataset. We believe we would achieve better performance using more images, but we choose to focus on the dataset construction and performance analyses as those are our main contributions.
> >
> > > "Could you explain the reasoning behind fig. 5 and why a direct comparison of model performance with DINO in absolute values was not included?"
> >
> > Please see our response to weakness #4
> > * We added a table that breaks down psychophysics performance by eccentricity in Figure 5
> > * We report a cross eccentricity summary because it is a standard way of evaluating psychophysics results
> >
> > > "Can you provide more context on potential real-world applications or implications of these results?"
> >
> > Please refer to our response to weakness #6.
> > * Peripheral vision guides human attention. It is already useful for recognition and could potentially aid performance in scene related tasks detection where peripheral vision plays a stronger role in performance
> > * By modeling peripheral vision, we could predict human visual behavior. This has applications in (1) driver safety where we could predict if a person sees a hazard, (2) content memorability where we could optimize images to capture attention, (3) UI/UX to create displays that easy to view, (4) foveated rendering, and (5) compression where peripheral vision could help models perform under reduced visual information.
> > * Finally, peripheral vision underlies a significant part of human visual representations. If we can successfully model it and human vision more generally, we will have learned a representation that is robust to many of the transformations that humans are robust to. Rather than training on a large set of perturbations, we could simply mimic a human visual representation that has already learned the desired robustness.
> > * We have expanded on real-world applications in the Discussion.

---

> > > ### Author Response · Authors · 2023-11-19
> > > **response part 3 (ethics)**
> > >
> > > > "any study aiming at modeling human behavior should include an analysis of potential harms"
> > >
> > > We thank reviewer eW1M for their concern around ethics. We value ethical research practices and aim to comply in every way.  Could reviewer eW1M clarify if the ethics concern is about larger implications for modeling human behavior or about ethical practices in human subject research?
> > >
> > > We included a statement about IRB protocol for human research in Appendix A.3.2, but we recognize that this deserves more explicit attention in the main body. To address this, we have added an ethics statement after the Discussion. It includes a note about possible harmful applications of modeling human behavior at a societal scale and further detail about the risks to human subjects in our study.

---

> ### Author Response · Authors · 2023-11-23
> **Additional Rebuttal**
>
> In addition to our earlier remarks, we now include a control human experiment where we measure detection performance directly on uniform TTM images (See Figure 4, and Appendix A.4). Our results show that performance on TTM images is similar to peripherally viewed images. These results validate that TTM adequately models the information loss in the human periphery on our specific psychophysics task. These results demonstrate that TTM is an interesting transformation to explore in DNNs as a validated model of peripheral vision. In addition, we emphasize that humans are able to perform the detection tasks on the transformed images, so it is worthwhile to evaluate DNNs in the same way. We appreciate your consideration.

---

### Official Review · Reviewer_xwTZ · 2023-11-01

**Soundness:** 3 good
**Presentation:** 3 good
**Contribution:** 2 fair
**Rating:** 6
**Confidence:** 4

**Summary:**

The paper introduces COCO-periph, a novel benchmark dataset tailored for advancing human-centric computer vision, building upon the well known  COCO dataset. This dataset is introduced to evaluate Convolutional Neural Networks' (CNNs) efficacy in recognizing peripheral objects. To create it they  utilize the Texture Tiling Model (TTM) to emulate the degradation of information with increasing object eccentricity.  CNNs’ considerably underperfom in this task, furthermore the behavior of models and humans is widely different. For instance,  in humans the drop in performance is gradual as the eccentricity augments, in contrast, CNNs drop sharpely once certain threshold is passed.

**Strengths:**

The paper proposes an interesting  benchmark that can bridge the gap or at least understand the limitations of CNNs.  I think it constitutes an excellent example of human-centered computer vision and its application in both understanding mechanisms to carry on this computations in humans and how models can increase performance if they are crafted with these effects in mind. The paper reads very well, and the authors provide a clear insight into what is the contribution of the paper.

**Weaknesses:**

I missed some references, that seems to be central for this problem, for instance (Zhang & Han ,2019) (Han, 2020). Perhaps linked to this the main missing point in the paper in my opinion was some baseline model that incorporates some of the mechanisms  or hypothesis from human vision that aim to solve this problem ( like EEN).

**Questions:**

I appreciate the authors' innovative approach with the COCO-periph dataset and the incorporation of the Texture Tiling Model (TTM) to simulate information loss in peripheral object recognition in the COCO dataset. However, I would like to request further elaboration on the decision not to present the TTM-generated images directly to human participants. My concern revolves around potential artifacts that could have been introduced during the experiments, possibly influencing the outcomes.

While previous results seem to corroborate a substantial match with human performance, it's crucial to note that the stimuli utilized in their study differ in certain aspects from those in previous research, such as the grayscale images presented in Ehinger & Rosenholtz (2016). Additionally, there seems to be a noticeable discrepancy in the texture quality of the images.

As well, although data-based solutions seems to shorten the distance  between CNNs and human performance,  I am eager to learn about the authors' insights and opinions regarding the fundamental mechanisms absent in CNNs but present in humans, which might account for these observed disparities. What specific aspects of human cognition and perception do the authors believe are not adequately captured by current CNN models, and how might future research bridge this gap to enhance the performance of computer vision systems?

---

> ### Author Response · Authors · 2023-11-19
> **response to review**
>
> We are grateful for reviewer xwTZ for their careful review of our paper, appreciation of our contributions, and suggestions for improvements that will strengthen our work.
>
> We thank the reviewer for pointing us to the additional references (Zhang and Han, 2019, and Han 2020), and have added these citations to the manuscript. We appreciate the suggestion to evaluate EEN which captures critical aspects of foveation and visual crowding. We choose to not evaluate this model and many other notable human-inspired DNNs such as VOneNet because they have not been built or trained to perform MS-COCO style object detection. In order to keep our analyses fair across models, we limited our work to models that met this criteria. We hope that our paper will inspire more work on human-inspired models for tasks like object detection so this becomes possible in the future.
>
> > "I would like to request further elaboration on the decision not to present the TTM-generated images directly to human participants."
>
> We agree that a strong comparison could have been to measure human performance directly on TTM images. We chose to prioritize original images in order to measure true peripheral performance. We believe this is more useful than TTM performance alone because our overall goal is to model peripheral vision. Although TTM is a strong model, we hope our work will enable better models of peripheral vision in the future. With original image performance, our psychophysics procedures can scale to new implementations of peripheral processing.
>
> That said, we recognize that our choice could have caused discrepancies in our comparisons. As additional controls, we included DNN psychophysics results for uniform, original, and pseudofoveated TTM (See AppendixFigures 14 & 15) to demonstrate that our results are robust to different variants of the transformation.
>
> > "it's crucial to note that the stimuli utilized in their study differ in certain aspects from those in previous research"
>
> We thank reviewer xwTZ for bringing up important limitations to consider in our work. On the concern about grayscale vs color, we acknowledge that color has not been widely tested in TTM. However, we chose to evaluate color images because the DNN models we tested were trained on color images. We did not want to disadvantage the models in that regard.
>
> We also agree that the synthesis process produces artifacts at times. We try to limit the impact of this by showing DNNs multiple seeds of the same image per eccentricity. We also argue that, despite this, TTM is still able to do a good job predicting performance and similar models pass metamer evaluations (Freeman & Simoncelli 2011). That said, we agree these limitations are important to consider, and we have added a note on limitations of TTM in the Discussion.
>
> > "I am eager to learn about the authors' insights and opinions regarding the fundamental mechanisms absent in CNNs but present in humans, which might account for these observed disparities"
>
>
> We thank reviewer xwTZ for their thoughtful question. TTM suggests that humans rely on a compressed set of summary statistics to represent much of the visual field. We propose that the problem of aligning DNNs with human perception is in some sense about learning those same statistics or features. Data is a very flexible way to achieve this, but it has limits as you noted. While the structure of the convolution operation has ties to human visual system function, the convolutional architecture of CNNs is distinct from the complex connectivity seen in the human visual cortex. These differences may also limit the ability of CNNs to perform on-par with their human counterparts. A next approach could be to train directly on TTM statistics themselves. Even better would be to optimize DNNs to learn a reduced set of features instead of over-parameterizing visual input as they currently do.
>
> To bridge the gap in object detection specifically, our work suggests that DNNs are trained to favor exact object location and size too much compared to humans. Relaxing this constraint could be key to aligning behavior.
>
> More broadly, we believe that task formulation is a critical area to explore in DNNs. TTM as a model suggests that one general representation can explain behavior on a variety of visual tasks. We believe an important future direction in bridging the gap between humans and DNNs is to optimize for generalization across a variety of tasks -- rather than maximizing for accuracy on a single task. Current benchmarks in computer vision do not encourage this, and we hope that our dataset and experiments can facilitate research in this direction.We have updated the Discussion to include some of these points.

---

> > ### Author Response · Authors · 2023-11-23
> > **additional control experiment and fovea-inspired model**
> >
> > To further address your concerns about testing more human-inspired models, we have added FoveaBox to our analyses. This architecture is loosely inspired by the fovea performing detection in an anchor-free manner. We find, however, that this model performs similar to RetinaNet on the psychophysics task (see Figure 4).
> >
> > Finally, in response to your question about directly measuring human performance on TTM images, we have added a control experiment doing exactly this. We measure human performance on the detection task using uniform TTM images for $3$ naive subjects and $2$ experts (See Figure 4). Our results show that performance on TTM images is similar to peripherally viewed images -- demonstrating that TTM adequately models the information loss in the human periphery on our specific psychophysics task. Thank you again for your remarks.

---

### Official Review · Reviewer_ogr4 · 2023-11-01

**Soundness:** 2 fair
**Presentation:** 2 fair
**Contribution:** 2 fair
**Rating:** 8
**Confidence:** 4

**Summary:**

This paper presents a new dataset generated by a modified version of the texture tiling model (TTM) to simulate the information available to human peripheral vision via image synthesis. The paper generates a new dataset, COCO-Periph, which is MS COCO processed through the TTM at different eccentricities. Humans performed a forced-choice object detection experiment, viewing the target object location at various eccentricities. Human accuracy was high well into the periphery (out to the 20 deg tested). Object detection models performing the same task using the TTM synthesised images showed performance drops to chance that occurred at smaller eccentricities than humans. Training on COCO-Periph raised this somewhat, but still not to human performance. The paper also presents some mixed results on noise robustness. The conclusions are that (1) the tested models are insufficient to match human peripheral performance (2) future model architectures should try to close this gap and (3) the COCO-Periph dataset is a valuable benchmark for achieving this.

**Strengths:**

1. The Pseudo-foveation is a clever way to produce computationally-tractable "fixated" images when the individual TTM eccentricities take so long to synthesise. The paper may consider citing Geisler and Perry [refs 1, 2], as they did this for blur.

2. The method to turn an object detection CNN into a 2IFC CNN is clever. It took me some time to understand the pooling of detection probabilities over the bounding box, but this is a nice way to compare the human and machine performance.

3. The 2IFC detection images created with inpainting is clever. I like the manipulation for creating target-absent images in a labelled object dataset.

### References


[1] J. S. Perry and W. S. Geisler, “Gaze-contingent real-time simulation of arbitrary visual fields,” in Proc. SPIE, Jun. 2002, pp. 57–69. doi: [https://doi.org/10.1117/12.469554](https://doi.org/10.1117/12.469554).

[2] W. S. Geisler and J. S. Perry, “Real-time simulation of arbitrary visual fields,” in Proceedings of the 2002 symposium on Eye tracking research & applications, ACM, 2002, pp. 83–87. Accessed: Nov. 11, 2014. [http://dl.acm.org/citation.cfm?id=507090](http://dl.acm.org/citation.cfm?id=507090)

**Weaknesses:**

### An alternative explanation for the results is that the TTM is an inadequate model of human peripheral vision

If the TTM discards more information than the human visual system in fact preserves, then we would expect this pattern of results: humans have better performance out to larger eccentricities than DNNs trained on COCO-Periph. This is not because of the model architecture, but because the COCO-Periph training set distorts images too much. If this were the case, then closing the gap between human and machine peripheral vision may not be about testing more advanced model architectures, but about finding a better representation for human peripheral vision.

The paper currently gives the impression that the TTM and related approaches have unequivocal support as a model of human peripheral vision:

> Humans viewing these transformed images perform visual tasks with an accuracy that well predicts human performance while fixating the original images

> This model has been tested to well predict human performance on an extensive number of behavioral tasks, including peripheral object recognition, visual search, and a variety of scene perception tasks

> rendered images that capture the amount of information available to humans at (5◦, 10◦, 15◦, 20◦)

> previous work demonstrated that TTM is a close match to human peripheral vision (Ehinger & Rosenholtz, 2016), so differences in performance cannot be explained by stimuli alone.

However, there is other work that would rather seem to support the alternative explanation above [refs 3, 4, 5, 6].
Under some circumstances, humans are more sensitive to image distortions than predicted by these texture models.
While there are undoubtedly tasks for which TTM does correlate well with human performance (e.g. visual search, scene gist recognition, etc), object detection as tested here may not be one of them.
Indeed, it is generally considered that crowding is about discrimination, not detection.
If the present paper had instead used an object discrimination task then the correlation between human and TTM-trained-DNNs might be much higher.
In any case, it remains uncertain whether the "gap in performance between humans and computer vision DNNs in the periphery" can or should be addressed through the development of more advanced DNN architectures and training on COCO-Periph (e.g. "In future, it would be interesting to train and evaluate transformer based- architectures.").
Rather, the adequacy of the TTM as a model of the human periphery for this specific purpose remains an open question.

### A more comprehensive description of the TTM should be provided

The paper lacks a comprehensive description of what the Texture Tiling Model (TTM) computes, which hinders a clear understanding of its workings.
The reference given for the original TTM is Rosenholtz et al., 2012a (e.g. Figure 8 caption).
That paper also does not contain a detailed description of what the TTM actually computes, that would be sufficient to re-implement the model.
It is also unclear whether the authors intend to release the TTM code, as the paper mentions the public release of the COCO-Periph dataset "along with all analysis code" but doesn't specify whether this includes the TTM.
For a more in-depth understanding of the TTM's summary statistics, reference to Portilla and Simoncelli (2000) is necessary, although this reference is absent from the paper and should be included.

### Data analysis and quantification

The absence of uncertainty indications in the data plots is a noticeable shortcoming.
Accuracies and psychometric function parameter estimates (Figure 3c, 5, 6, etc) should give indications of uncertainty in the data and parameter estimates (e.g. confidence intervals).
Are the psychometric function parameters estimated from all human data pooled?
For each participant individually?
Otherwise it's hard to evaluate the precision of the data.

For instance, Figure 5 shows human criticial eccentricity $\mu$ values of around 22.
However, if this is a fit to the accuracy data in Figure 3c (as it seems to be) then this value seems implausible: the psychometric function threshold is likely far higher.
This point seems to be supported by the fact that the two images that were removed (A. 3.4, Figure 13) appear to show basically no eccentricity effect -- including these images would further support a far higher $\mu$ value.

Moreover, the step size of TTM eccentricities might be insufficient to offer robust constraints on the model's $\mu$ values.
Figure 3c illustrates multiple almost-equally-plausible psychometric curves that could be drawn between the relevant points of DINO FocalNet, potentially altering the $\mu$ estimates. These issues highlight the importance of disclosing uncertainty in parameter estimates, making it difficult to evaluate the improvement gained from training or fine-tuning on COCO-Periph.


### References


[3] T. S. A. Wallis, M. Bethge, and F. A. Wichmann, “Testing models of peripheral encoding using metamerism in an oddity paradigm,” Journal of Vision, vol. 16, no. 2, p. 4, Mar. 2016, doi: [https://doi.org/10.1167/16.2.4](https://doi.org/10.1167/16.2.4).

[4] A. Deza, A. Jonnalagadda, and M. P. Eckstein, “Towards metamerism via foveated style transfer,” ICLR 2019. [https://openreview.net/forum?id=BJzbG20cFQ](https://openreview.net/forum?id=BJzbG20cFQ)

[5] T. S. A. Wallis, C. M. Funke, A. S. Ecker, L. A. Gatys, F. A. Wichmann, and M. Bethge, “Image content is more important than Bouma’s Law for scene metamers,” eLife, vol. 8, p. e42512, Apr. 2019, doi: [https://doi.org/10.7554/eLife.42512](https://doi.org/10.7554/eLife.42512).

[6] W. F. Broderick, G. Rufo, J. Winawer, and E. P. Simoncelli, “Foveated metamers of the early visual system”. bioRxiv. doi: [https://doi.org/10.1101/2023.05.18.541306](https://doi.org/10.1101/2023.05.18.541306)

**Questions:**

1. Is there something I have overlooked that rules out the alternative explanation for the results proposed above?

2. How often does `if pprob = aprob then acc = acc + 0.5` occur?

3. Is it possible that training on detection makes the model worse at the 2IFC task for some images? e.g. if a single confident object detection yields a lower summed "probability" (`pprob` and `aprob`) than multiple less-confident detections that overlap the padded box?

---

> ### Author Response · Authors · 2023-11-19
> **Response to Reviewer #1**
>
> We thank reviewer ogr4 for the thoughtful and careful review of our paper, appreciation of the strengths and contributions of our work, and for many helpful suggestions that will improve the paper’s impact.
>
> Firstly, we thank the reviewer for pointing us to the Perry & Geisler citations which stitch together blur images for computationally-tractable fixation, similar to what we propose using uniform TTM images. We have included these citations in the manuscript.
>
> >“An alternative explanation for the results is that the TTM is an inadequate model of human peripheral vision”
>
> We appreciate reviewer ogr4’s thoughtful alternative explanation for our results due to TTM throwing away too much information. Indeed, like all peripheral vision models,  TTM does indeed underpredict human performance for certain stimuli and situations. Among all the peripheral models currently available, we chose TTM to create our dataset because it has been validated on the widest range of tasks compared to any similar texture-based model. It is true that the object detection task tested here differs slightly from the object detection task tested in Ehinger & Rosenholtz 2016, with more hallmarks of a discrimination task. We note that models sharing pooled TTM statistics have been widely tested on discrimination with the metamer paradigm (Freeman & Simoncelli 2011, Wallis 2019). We agree with reviewer ogr4 that there are newer models of peripheral vision that each contribute their own improvements such as style-transfer based models like (Wallis 2019, Deza 2017)). However these models focus primarily on the constraint of metamerism rather than predicting task performance which we believe is important and interesting to consider when making comparisons to DNNs. In our work, although we use object detection as an exemplar, our goal is for COCO-Periph to be applied in many other visual tasks such as search which is labeled in COCO (Chen et al., 2021), as well as scene and image recognition which could be explored using the COCO things and stuff labels (Caesar et al., 2018). We appreciate that our language did not discuss the limitations of TTM and other related models, and we have updated our manuscript to address these points.
>
> In weighing the reviewer’s hypothesis against that presented in the paper, we acknowledge that the limitations of current state of the art peripheral vision models like TTM could contribute in part to our results. We see this as an important point to share with readers, and have incorporated this into the updated manuscript. We argue however, that the limitations of TTM cannot completely explain our results for the following reasons:
>
> 1. While TTM throwing away too much information could create a ceiling on model performance as compared to humans, our results also demonstrate the brittle falloff of performance for machine models as compared to the gradual falloff of performance in humans (Figure 3c). TTM uses the same statistical parameters at all eccentricities, modifying only pooling region size, which should not cause a sharp performance drop between low and high eccentricities. Indeed, we show when fine-tuning models on single eccentricities boosts performance at other eccentricities (Appendix Figure 22). Therefore, it is unlikely that this sharp performance drop off seen is attributable to TTM itself, and much more likely to be TTM-transforms being out of distribution to untrained models.
>
> 2. The differences in performance between the different model architectures tested (ie DINO, RCNN, DETR etc) cannot be explained by loss of information in TTM, as these models received the same TTM-transformed input images.
>
> 3. Relatedly, the improvement in performance seen after training and fine-tuning models on COCO-Periph demonstrates a change in the ability of even the same model architecture to process peripherally-transformed images differently. Again, the baseline, fine-tuned, and trained models receive the same TTM-transformed input images.
>
> 4. Also related, the improvement in corruption robustness after fine-tuning and training with COCO-Periph is orthogonal to any limitations of TTM as a peripheral vision model.
>
> 5. Furthermore, our results in comparing model performance in detection vs recognition are likewise orthogonal to any limitations of TTM as a peripheral vision model.

---

> > ### Comment · Reviewer_ogr4 · 2023-11-22
> >
> > In my view, none of the points 1-4 adequately address the core issue.
> > My alternative explanation asserts that no model will be able to match human performance at this task when using TTM images (when humans perform the task using their peripheral vision).
> > The reason is because there is insufficient information in the TTM images at larger eccentricities to perform the task at human level.
> >
> > I think the necessary experiment to check whether this explanation holds would be to have humans do the task viewing the 15 and 20 degree TTM images with their central vision: can they reach the same or better performance as they do with their own peripheral vision?
> > (Reviewer xwTZ asked about this in their review).
> > If this were the case, it would show that there is indeed sufficient information preserved in the TTM-representation, and that it is worthwhile to study new models utilising COCO-periph to bring us closer to a human-like model.
> > If on the other hand, humans are no better than the current best model at 15 and 20 degrees, this would be consistent with the COCO-Periph images being too distorted.
> >
> > The authors cite a previous paper that has basically done this experiment (Ehinger and Rosenholtz, 2016) as
> >
> > > previous work demonstrated that TTM is a close match to human peripheral vision, including for object detection tasks (Ehinger & Rosenholtz, 2016).
> >
> > The data in that paper (their Figure 5) does not convincingly demonstrate what the authors write above.
> > In fact, the accuracies for the five object detection images in that paper are all lower for TTM images than for fixating in the real images: (fixation, TTM classification) are approximately (0.76, 0.55), (0.76, 0.6), (0.74, 0.62), (0.7, 0.64) and (0.67, 0.58).
> > That's remarkably similar to the accuracy drop between humans and models in the present data (new version Figure 4 at 20 deg: human performance at about 0.75, best model at 0.63).
> > If this relationship held in the present stimuli and tasks then it would be consistent with the explanation above.
> > So repeating this experiment with the current stimuli and tasks would be the most convincing test.
> > If the results hold, it would suggest that models can't be expected to ever meet human peripheral performance on object detection using TTM images, because too much information is lost. (shortcut learning notwithstanding).
> >
> > To the other points:
> >
> > 1. It's true that the set of statistics is the same, but I think it's difficult to say how the computation of these statistics over different region sizes will interact with target object size in image, model receptive field sizes, and so on. I think it's quite plausible that this could cause a sharp falloff in performance.
> >
> > 2. The potential loss of information in the TTM is only relevant to the comparison to human performance. It's definitely plausible that there could additionally be differences in performance for different models, but this is irrelevant to the question of whether it's human-like.
> >
> > 3 + 4. Model improvements after training on noisier / corrupted images are understandable within the context of becoming more robust / generalise better to out of domain images, but this again doesn't address the core issue.
> >
> > Whether or not this alternative explanation is correct, we come back to the main contribution of this work, which is the new dataset.
> > I do agree with the authors' new sentences in their conclusion, that this dataset may help to make some progress in various applications.
> > However, my view in the end is that the broad usefulness of this dataset for developing more human-like models has not been adequately demonstrated in this paper.

---

> > > ### Author Response · Authors · 2023-11-23
> > > **Response to Request for Additional Experiment**
> > >
> > > # Control Experiment
> > >
> > > We thank reviewer ogr4 for their additional feedback. In response, we have included results in our updated manuscript for a control experiment where we test humans on our same object detection task with the same images transformed with uniform TTM (See Appendix Section A.4). For $5$ subjects ($2$ expert, and $3$ naive), where we show that detection performance is on par with original images. We report results for both naive subjects only, as well as all 5 subjects together. This experiment is precisely that requested by the reviewer, and we include in addition eccentricities $5$ and $10$, along with the eccentricities $15$ and $20$ that the reviewer requested. We also report per-image accuracy for this control experiment in FIgure 13. This control experiment demonstrates that our DNN results cannot be explained by TTM throwing away too much information.
> > >
> > > # Response to questions in initial review
> > >
> > > In addition, we have also updated the manuscript to address some outstanding issues from the original comment:
> > >
> > > > How often does if pprob = aprob then acc = acc + 0.5 occur?
> > >
> > > We investigated how often aprob equals pprob and found that for the baseline R50-RCNN model, this represents only cases where the prediction probability is zero (no predictions above the 1% threshold) in both the present and absent mongrel pair. Over the image set, it occurs at least once over the 100 present/absent pairings, in 10/26 images, and for each eccentricity (80$^\circ$,160$^\circ$,240$^\circ$,320$^\circ$), occurs (70,286,357,239) times over the 2600 pairings per eccentricity, for a total of (2.7%,11%,13.7%,9.2%) overall. For the baseline RCNN, we observe that aprob never equals pprob because the model had equivalent predictions in the present and absent images. This only occurs when no objects are predicted in either image. We have added a note on this in Appendix Section A5.
> > >
> > > > Is it possible that training on detection makes the model worse at the 2IFC task for some images? e.g. if a single confident object detection yields a lower summed "probability" (pprob and aprob) than multiple less-confident detections that overlap the padded box?
> > >
> > > To address your concern about summed probabilities, we re-analyzed the RCNN models taking the max probability rather than the summed probability of all proposed bounding boxes (see Figure 26). Taking the max instead of the sum actually *reduces* DNN performance on the psychophysics task. This suggests that the scenario where summing disadvantages a single confident prediction does not happen frequently. Rather, it is more favorable to take the approach that we report in our main results.
> > >
> > > In addition we would like to add that figure 5 gives more insight to the question about the possibility of a single confident detection yielding a lower sum than many few detections. Here, we separate out the machine psychophysics task into recognition vs detection by passing the ground truth bounding box to the model, allowing us to disentangle localization and recognition. In effect, this forces the model to calculate one prediction probability for a single, correct bounding box. We see that for both the baseline and the fine tuned models, this results in similar performance, implying that for these models, multiple low-probability bounding boxes are likely not outweighing a single high-probability detection. If this were the case, we would expect improved performance in 5b as compared to 5a. Furthermore, for the trained from scratch model, enforcing a single guess results in worse performance, indicating that weighing the summation of multiple predictions helps the model, rather than hinders it. In summary, while it is possible that this could occur at some times, evidence points to this being a small effect, if it occurs at all.

---

> ### Author Response · Authors · 2023-11-19
> **Response to Reviewer #1 (Part 2)**
>
> >“A more comprehensive description of the TTM should be provided”
>
> We are thankful to the reviewer for asking for clarification regarding the release of our code, and for a more comprehensive description of TTM in our paper. Indeed, as authors we strongly value reproducibility in our work and aim to support this wherever possible. We have updated the manuscript to clarify that we will release both our dataset, as well as the code for the uniform TTM model used to create the dataset, in addition to the machine psychophysics model. Furthermore, we have supplemented the citation to TTM with a citation to the original TTM codebase (https://dspace.mit.edu/handle/1721.1/124819), as we aim to ensure full transparency. We especially appreciate the note on adding the reference to Portilla and Simoncelli, which we have now included in the paper.
>
> >“Data analysis and quantification”
>
> We appreciate the reviewer asking for clarification regarding how data is pooled in figures 3c, 5, and 6 - we have added explanations for this in each of these figure captions. To clarify, figure 3 shows data for all human subjects and models for a single image. Figure 5 is not a fit to only the data from figure 3, but averages data over all human subjects over all images. We thank the reviewer for pointing out the absence of error bars for our critical eccentricity (\mu) fits in figures 3c, 5, and 6. For figure 3c which shows results for an individual image, this is indeed possible for the human data, and we have included these, as well as explanation in the figure caption. As figures 5 and 6 show data averaged over all human subjects and images, with different images having different inherent difficulties (object size, amount of crowding, contrast, etc - see Figure 7), error bars in these plots would not reflect the overall variability in performance in a meaningful way.
>
> Regarding reporting uncertainty with error bars for the model data, we note that measuring uncertainty even for a single image (figure 3c) is not possible for the models given the machine psychophysics, which averages the simulated response of a single model over 100 trials (100 possible pairs of 10 present, 10 absent mongrels) at each eccentricity to give each accuracy value. There is no sense of multiple ‘individuals’ of the same model, and models are deterministic, always giving the same response for a given present/absent pair. We previously considered splitting the 100 trials into a simulated 10 accuracy values each with 10 trials to allow for the inclusion of error analysis, but found that calculating an accuracy value over only 10 trials gave too little accuracy granularity in the reported accuracy values (i.e. there would only be 10 possible accuracy values).
>
> In addition, we value the reviewer’s concerns regarding the psychometric function fits to accuracy data. We have added clarification around Figure 5 being an average over fits to all images. Regarding the concern about removing two images from the dataset (Figure 13), we agree that including these images would have slightly raised the estimate of human \mu overall. We note that this would have strengthened our finding that humans outperform models, but nonetheless, being unable to fit a psychometric curve to this data we were forced to remove them in order to fairly compare humans and machines with a psychometric fit.  To address all of these concerns, we have included a second plot in Figure 5, which represents accuracy data for both humans and models before any function fits or removal of images for which fitting failed.
>
> Relatedly, we share the reviewer’s desire to evaluate these models at a wider range of eccentricities and with smaller step-sizes. Put simply, the computational requirements of creating these mongrel images (1-2 hours per image for the entire 328,000 MS-COCO dataset) limits the number of eccentricities we can feasibly explore.
>
> To address these above concerns about psychometric function fits specifically, and along the lines of the suggestions by reviewer eW1M, we have included a plot reporting averaged accuracy results for each model, before the psychometric fit (Figure 5). This demonstrates the falloff in performance in machines happening more sharply and at earlier eccentricities as compared to humans.

---

> ### Author Response · Authors · 2023-11-19
> **Response to Reviewer #1 (Part 3)**
>
> > Question #1: "Is there something I have overlooked that rules out the alternative explanation for the results proposed above?"
>
> Please see our response above. In short, we agree that this alternative explanation could impose a ceiling on the machine results as compared to human, but due to the evidence listed this explanation cannot account for the entirety of the results. We appreciate the reviewer offering this insight, and have included this in the discussion.
>
> > Question #2 "How often does if pprob = aprob then acc = acc + 0.5 occur?"
>
> We designed this as a catch for a potential unlikely edge-case, and do not expect that is occurs often, but we will investigate this further.
>
> > Question #3: "Is it possible that training on detection makes the model worse at the 2IFC task for some images? e.g. if a single confident object detection yields a lower summed "probability" (pprob and aprob) than multiple less-confident detections that overlap the padded box?"
>
> We appreciate the reviewer bringing up this point, as we went to great lengths in designing the machine 2IFC task to evaluate models as fairly as possible. We decided to utilize a summed probability rather than the maximum probability (best prediction) because in our observations, we found that baseline models tended to make few or no predictions when no objects were present, rather than many poor ones (Figure 1, top). In addition, when objects were present, models tended to make multiple overlapping predictions (Figure 1, bottom). Rather than enforce a match on object label and exact location, we wanted to give models the best chance at solving the present/absent tasks, as human participants may have been able to guess by selecting the more ‘object-y’ image. While it is possible that training could have made performance worse in some cases such as that aptly pointed out by the reviewer, we found that evaluating the sum of probabilities rather than the highest probability resulted in better model performance overall. We have included a list of design choices for the machine psychophysics in the Appendix Section A4.

---

### Meta-Review · Area_Chair_eZ32 · 2023-12-05

**Metareview:**

The paper studies the gap between human vision and machine vision. The authors modify the texture tiling model (TTM) to generate a version of the popular COCO benchmark called COCO-Periph. COCO-Periph consists of transformed images that contain the information captured by peripheral human vision. The authors benchmark popular machine vision models and compare them to human performance. They show that machine vision systems do not model peripheral vision like humans and points to a clear way for improvement.
The author’s response also addressed concerns by the reviewers who raised their score. I also believe that the author’s responses address the concerns of RxwTZ.

**Justification For Why Not Higher Score:**

This paper shows a clear gap between machine vision and human vision on the COCO-Periph benchmark. The impact of this work would be greater if the authors to connect this observation with any of the following (a) other datasets; (b) practical applications where this matters; (c) specific architectural or training issues that lead to this gap.

**Justification For Why Not Lower Score:**

Clear contribution in identifying a gap with many modern machine vision systems. This work will likely help many future methods.

---

### Decision · Program_Chairs · 2024-01-16

Accept (poster)